# HUMAN-LEVEL ATARI 200X FASTER

**Steven Kapturowski, Víctor Campos\*, Ray Jiang\*, Nemanja Rakićević, Hado van Hasselt, Charles Blundell, Adrià Puigdomènech Badia**
DeepMind, \*Equal contribution
`{skapturowski,camunez,rayjiang,rakicevic,hado,`
`cblundell,adriap}@deepmind.com`

## ABSTRACT

The task of building general agents that perform well over a wide range of tasks has been an important goal in reinforcement learning since its inception. The problem has been subject of research of a large body of work, with performance frequently measured by observing scores over the wide range of environments contained in the Atari 57 benchmark. Agent57 was the first agent to surpass the human benchmark on all 57 games, but this came at the cost of poor data-efficiency, requiring nearly 80 billion frames of experience to achieve. Taking Agent57 as a starting point, we employ a diverse set of strategies to achieve a 200-fold reduction of experience needed for all games to outperform the human baseline within our novel agent MEME. We investigate a range of instabilities and bottlenecks we encountered while reducing the data regime, and propose effective solutions to build a more robust and efficient agent. We also demonstrate competitive performance with high-performing methods such as Muesli and MuZero. Our contributions aim to achieve faster propagation of learning signals related to rare events, stabilize learning under differing value scales, improve the neural network architecture, and make updates more robust under a rapidly-changing policy.

## 1 INTRODUCTION

To develop generally capable agents, the question of how to evaluate them is paramount. The Arcade Learning Environment (ALE) (Bellemare et al., 2013) was introduced as a benchmark to evaluate agents on an diverse set of tasks which are interesting to humans, and developed externally to the Reinforcement Learning (RL) community. As a result, several games exhibit reward structures which are highly adversarial to many popular algorithms. Mean and median human normalized scores (HNS) (Mnih et al., 2015) over all games in the ALE have become standard metrics for evaluating deep RL agents. Recent progress has allowed state-of-the-art algorithms to greatly exceed average human-level performance on a large fraction of the games (Van Hasselt et al., 2016; Espeholt et al., 2018; Schrittwieser et al., 2020). However, it has been argued that mean or median HNS might not be well suited to assess generality because they tend to ignore the tails of the distribution (Badia et al., 2019). Indeed, most state-of-the-art algorithms achieve very high scores by performing very well on most games, but completely fail to learn on a small number of them.

Agent57 (Badia et al., 2020) was the first algorithm to obtain above human-average scores on all 57 Atari games. However, such generality came at the cost of data efficiency; requiring tens of billions of environment interactions to achieve above average-human performance in some games, reaching a figure of 78 billion frames before beating the human benchmark in all games. Data efficiency remains a desirable property for agents to possess, as many real-world challenges are data-limited by time and cost constraints (Dulac-Arnold et al., 2019). In this work, we develop an agent that is as general as Agent57 but that requires only a fraction of the environment interactions to achieve the same result.

There exist two main trends in the literature when it comes to measuring improvements in the learning capabilities of agents. One approach consists in measuring performance after a limited budget of interactions with the environment. While this type of evaluation has led to important progress (Espeholt et al., 2018; van Hasselt et al., 2019; Hessel et al., 2021), it tends to disregard problems which are considered too hard to be solved within the allowed budget (Kaiser et al., 2019). On the other hand, one can aim to achieve a target end-performance with as few interactions as

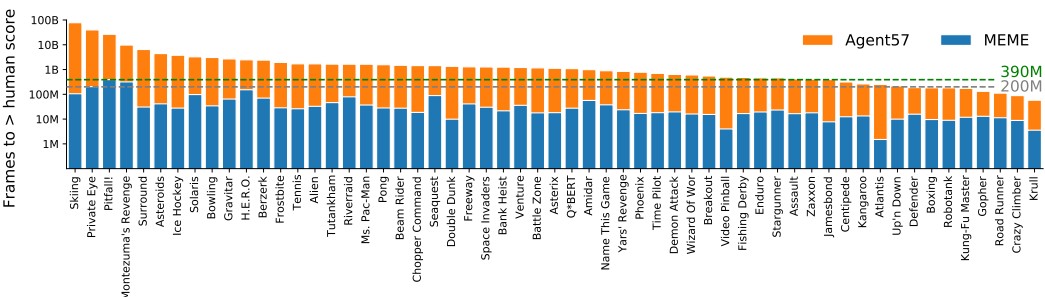

Figure 1: Number of environment frames required by agents to outperform the human baseline on each game (in log-scale). Lower is better. On average, MEME achieves above human scores using $62\times$ fewer environment interactions than Agent57. The smallest improvement is $10\times$ (*Road Runner*), the maximum is $734\times$ (*Skiing*), and the median across the suite is $36\times$. We observe small variance across seeds (c.f. Figure 8).

possible (Silver et al., 2017; 2018; Schmitt et al., 2020). Since our goal is to show that our new agent is as general as Agent57, while being more data-efficient, we focus on the latter approach.

Our contributions can be summarized as follows. Building off Agent57, we carefully examine bottlenecks which slow down learning and address instabilities that arise when these bottlenecks are removed. We propose a novel agent that we call MEME, for *MEME is an Efficient Memory-based Exploration agent*, which introduces solutions to enable taking advantage of three approaches that would otherwise lead to instabilities: training the value functions of the whole family of policies from Agent57 in parallel, on all policies' transitions (instead of just the behaviour policy transitions), bootstrapping from the online network, and using high replay ratios. These solutions include carefully normalising value functions with differing scales, as well as replacing the Retrace update target (Munos et al., 2016) with a soft variant of Watkins' Q($\lambda$) (Watkins & Dayan, 1992) that enables faster signal propagation by performing less aggressive trace-cutting, and introducing a trust-region for value updates. Moreover, we explore several recent advances in deep learning and determine which of them are beneficial for non-stationary problems like the ones considered in this work. Finally, we examine approaches to robustify performance by introducing a policy distillation mechanism that learns a policy head based on the actions obtained from the value network without being sensitive to value magnitudes. Our agent outperforms the human baseline across all 57 Atari games in 390M frames, using two orders of magnitude fewer interactions with the environment than Agent57 as shown in Figure 1.

## 2 RELATED WORK

Large scale distributed agents have exhibited compelling results in recent years. Actor-critic (Espeholt et al., 2018; Song et al., 2020) as well as value-based agents (Horgan et al., 2018; Kapturowski et al., 2018) demonstrated strong performance in a wide-range of environments, including the Atari 57 benchmark. Moreover, approaches such as evolutionary strategies (Salimans et al., 2017) and large scale genetic algorithms (Such et al., 2017) presented alternative learning algorithms that achieve competitive results on Atari. Finally, search-augmented distributed agents (Schrittwieser et al., 2020; Hessel et al., 2021) also hold high performance across many different tasks, and concretely they hold the highest mean and median human normalized scores over the 57 Atari games. However, all these methods show the same failure mode: they perform poorly in hard exploration games, such as *Pitfall!*, and *Montezuma's Revenge*. In contrast, Agent57 (Badia et al., 2020) surpassed the human benchmark on all 57 games, showing better general performance. Go-Explore (Ecoffet et al., 2021) similarly achieved such general performance, by relying on coarse-grained state representations via a downscaling function that is highly specific to Atari.

Learning as much as possible from previous experience is key for data efficiency. Since it is often desirable for approximate methods to make small updates to the policy (Kakade & Langford, 2002; Schulman et al., 2015), approaches have been proposed for enabling multiple learning steps over the same batch of experience in policy gradient methods to avoid collecting new transitions for

every learning step (Schulman et al., 2017). This decoupling between collecting experience and learning occurs naturally in off-policy learning agents with experience replay (Lin, 1992; Mnih et al., 2015) and Fitted Q Iteration methods (Ernst et al., 2005; Riedmiller, 2005). Multiple approaches for making more efficient use of a replay buffer have been proposed, including prioritized sampling of transitions (Schaul et al., 2016), sharing experience across populations of agents (Schmitt et al., 2020), learning multiple policies in parallel from a single stream of experience (Riedmiller et al., 2018), or reanalyzing old trajectories with the most recent version of a learned model to generate new targets in model-based settings (Schrittwieser et al., 2020; 2021) or to re-evaluate goals (Andrychowicz et al., 2017).

The ATARI100k benchmark (Kaiser et al., 2019) was introduced to observe progress in improving the data efficiency of reinforcement learning agents, by evaluating game scores after 100k agent steps (400k frames). Work on this benchmark has focused on leveraging the use of models (Ye et al., 2021; Kaiser et al., 2019; Long et al., 2022), unsupervised learning (Hansen et al., 2019; Schwarzer et al., 2021; Srinivas et al., 2020; Liu & Abbeel, 2021), or greater use of replay data (van Hasselt et al., 2019; Kielak, 2020) or augmentations (Kostrikov et al., 2021; Schwarzer et al., 2021). While we consider this to be an important line of research, this tight budget produces an incentive to focus on a subset of games where exploration is easier, and it is unclear some games can be solved from scratch with such a small data budget. Such a setting is likely to prevent any meaningful learning on hard-exploration games, which is in contrast with the goal of our work.

## 3 BACKGROUND: AGENT57

Our work builds on top of Agent57, which combines three main ideas: (*i*) a distributed deep RL framework based on Recurrent Replay Distributed DQN (R2D2) (Kapturowski et al., 2018), (*ii*) exploration with a family of policies and the Never Give Up (NGU) intrinsic reward (Badia et al., 2019), and (*iii*) a meta-controller that dynamically adjusts the discount factor and balances exploration and exploitation during training, by selecting from a family of policies. Below, we give a general introduction to the problem setting and some of the relevant components of Agent57.

**Problem definition.** We consider the problem of discounted infinite-horizon RL in Markov Decision Processes (MDP) (Puterman, 1994). The goal is to find a policy $\pi$ that maximises the expected sum of discounted future rewards, $\mathbb{E}_\pi[\sum_{t \geq 0} \gamma^t r_t]$, where $\gamma \in [0, 1)$ is the discount factor, $r_t = r(x_t, a_t)$ is the reward at time $t$, $x_t$ is the state at time t, and $a_t \sim \pi(a|x_t)$ is the action generated by following some policy $\pi$. In the off-policy learning setting, data generated by a behavior policy $\mu$ is used to learn about the target policy $\pi$. This can be achieved by employing a variant of Q-learning (Watkins & Dayan, 1992) to estimate the action-value function, $Q^\pi(x, a) = \mathbb{E}_\pi[\sum_{t \geq 0} \gamma^t r_t | x_t = x, a_t = a]$. The estimated action-value function can then be used to derive a new policy $\pi(a|x)$ using the $\epsilon$-greedy operator $\mathcal{G}_\epsilon$ (Sutton & Barto, 2018) [1]. This new policy can then be used as the target policy for another iteration, repeating the process. Agent57 uses a deep neural network with parameters $\theta$ to estimate action-value functions, $Q^\pi(x, a; \theta)$ [2], trained on return estimates $G_t$ derived with Retrace from sequences of off-policy data (Munos et al., 2016). In order to stabilize learning, a target network is used for bootstrapping the return estimates using double Q-learning (Van Hasselt et al., 2016); the parameters of this target network, $\theta_T$, are periodically copied from the online network parameters $\theta$ (Mnih et al., 2015). Finally, a value-function transformation is used to compress the wide range of reward scales present in Atari, as in (Pohlen et al., 2018).

**Distributed RL framework.** Agent57 is a distributed deep RL agent based on R2D2 that decouples acting from learning. Multiple actors interact with independent copies of the environment and feed trajectories to a central replay buffer. A separate learning process obtains trajectories from this buffer using prioritized sampling and updates the neural network parameters to predict action-values at each state. Actors obtain parameters from the learner periodically. See Appendix E for more details.

**Exploration with NGU.** Agent57 uses the Never Give Up (NGU) intrinsic reward to encourage exploration. It aims at learning a family of $N = 32$ policies which maximize different weightings of the extrinsic reward given by the environment ($r_t^e$) and the intrinsic reward ($r_t^i$), $r_{j,t} = r_t^e + \beta_j r_t^i$ ($\beta_j \in \mathbb{R}^+, j \in \{0, \dots, N-1\}$). The value of $\beta_j$ controls the degree of exploration, with higher values

---

[1] We also use $\mathcal{G} := \mathcal{G}_0$ to denote the pure greedy operator ($\epsilon = 0$).

[2] For convenience, we occasionally omit $(x, a)$ or $\theta$ from $Q(x, a; \theta)$, $\pi(a|x; \theta)$ when it is unambiguous.

Figure 2: MEME agent network architecture. The output of the LSTM block is passed to each of the N members of the family of policies, depicted as a light-grey box. Each policy consists of an Q-value and policy heads. The Q-value head is similar as in Agent57 paper, while the policy head is introduced for acting and target computation, and trained via policy distillation.

encouraging more exploratory behaviors, and each policy in the family is optimized with a different discount factor $\gamma_j$. The Universal Value Function Approximators (UVFA) framework (Schaul et al., 2015) is employed to efficiently learn $Q^{\pi^j}(x, a; \theta) = \mathbb{E}_{\pi^j}[\sum_{t \geq 0} \gamma_j^t r_{j,t} | x_t = x, a_t = a]$ (we use a shorthand notation $Q^j(x, a; \theta)$) for $j \in \{0, \ldots, N-1\}$ using a single set of shared parameters $\theta$. The policy $\pi^j(a|x)$ can then be derived using the $\epsilon$-greedy operator as $\mathcal{G}_\epsilon Q^j(x, a; \theta)$. We refer the reader to Appendix H for more details.

**Meta-controller.** Agent57 introduces an adaptive meta-controller that decides which policies from the family of N policies to use for collecting experience based on their episodic returns. This naturally creates a curriculum over $\beta_j$ and $\gamma_j$ by adapting their value throughout training. The optimization process is formulated as a non-stationary bandit problem. A detailed description about the meta-controller implementation is provided in Appendix F.

**Q-function separation.** The architecture of the Q-function in Agent57 is implemented as two separate networks in order to split the intrinsic and extrinsic components. The network parameters of $Q^j(x, a; \theta_e)$ and $Q^j(x, a; \theta_i)$ are separate and independently optimized with $r_j^e$ and $r_j^i$, respectively. The main motivation behind this decomposition is to allow each network to adapt to the scale and variance associated with their corresponding reward, as well as preventing the gradients of the decomposed intrinsic and extrinsic value function heads from interfering with each other.

## 4 MEME: IMPROVING THE DATA EFFICIENCY OF AGENT57

This section describes the main algorithmic contributions of the MEME agent, aimed at improving the data efficiency of Agent57. These contributions aim to achieve faster propagation of learning signals related to rare events (**A**), stabilize learning under differing value scales (**B**), improve the neural network architecture (**C**), and make updates more robust under a rapidly-changing policy (**D**). For clarity of exposition, we label the contributions according to the type of limitation they address.

**A1 Bootstrapping with online network.** Target networks are frequently used in conjunction with value-based agents due to their stabilizing effect when learning from off-policy data (Mnih et al., 2015; Van Hasselt et al., 2016). This design choice places a fundamental restriction on how quickly changes in the Q-function are able to propagate. This issue can be mitigated to some extent by simply updating the target network more frequently, but the result is typically a less stable agent. To accelerate signal propagation while maintaining stability, we use online network bootstrapping, and we stabilize the learning by introducing an approximate trust region for value updates that allows us to filter which samples contribute to the loss. The trust region masks out the loss at any timestep for which *both* of the following conditions hold:

$$|Q^j(x_t, a_t; \theta) - Q^j(x_t, a_t; \theta_T)| > \alpha \sigma_j \tag{1}$$

$$\text{sgn}(Q^j(x_t, a_t; \theta) - Q^j(x_t, a_t; \theta_T)) \neq \text{sgn}(Q^j(x_t, a_t; \theta) - G_t) \tag{2}$$

where $\alpha$ is a fixed hyperparameter, $G_t$ denotes the return estimate, $\theta$ and $\theta_T$ denote the online and target parameters respectively, and $\sigma_j$ is the standard deviation of the TD-errors (a more precise description of which we defer until **B1**). Intuitively, we only mask if *the current value of the online network is outside of the trust region* (Equation 1) *and the sign of the TD-error points away*

*from the trust region* (Equation 2), as depicted in Figure 3 in red. We note that a very similar trust region scheme is used for the value-function in most Proximal Policy Optimization (PPO) implementations (Schulman et al., 2017), though not described in the original paper. In contrast, the PPO version instead uses a constant threshold, and thus is not able to adapt to differing scales of value functions.

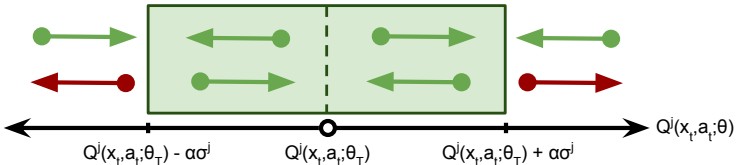

Figure 3: Trust region. The position of dots is given by the relationship between the values predicted by the online network, $Q^j(x_t, a_t; \theta)$, and the values predicted by the target network, $Q^j(x_t, a_t; \theta_T)$ (Equation 1 and left hand side of Equation 2), the box represents the trust region bounds defined in Equation 1, and the direction of the arrow is given by the right hand side of Equation 2. Green-colored transitions are used in the loss computation, whereas red ones are masked out.

**Ⓐ2 Target computation with tolerance.** Agent57 uses Retrace (Munos et al., 2016) to compute return estimates from off-policy data, but we observed that it tends to cut traces too aggressively when using $\epsilon$-greedy policies thus slowing down the propagation of information into the value function. Preliminary experiments showed that data efficiency was improved in many dense-reward tasks when replacing Retrace with Peng's Q($\lambda$) (Peng & Williams, 1994), but its lack of off-policy corrections tends to result in degraded performance as data becomes more off-policy (e.g. by increasing the expected number of times that a sequence is sampled from replay, or by sharing data across a family of policies). This motivates us to propose an alternative return estimator, which we derive from Q($\lambda$) (Watkins & Dayan, 1992):

$$G_t = \max_a Q(x_t, a) + \sum_{k \geq 0} \left( \prod_{i=0}^{k-1} \lambda_i \right) \gamma^k (r_{t+k} + \gamma \max_a Q(x_{t+k+1}, a) - \max_a Q(x_{t+k}, a)) \quad (3)$$

where $\prod_{i=0}^{k-1} \lambda_i \in [0, 1]$ effectively controls how much information from the future is used in the return estimation and is generally used as a trace cutting coefficient to perform off-policy correction. Peng's Q($\lambda$) does not perform any kind of off-policy correction and sets $\lambda_i = \lambda$, whereas Watkins' Q($\lambda$) (Watkins & Dayan, 1992) aggressively cuts traces whenever it encounters an off-policy action by using $\lambda_i = \lambda \mathbb{I}_{a_i \in \mathrm{argmax}_a Q(x_i, a)}$, where $\mathbb{I}$ denotes the indicator function. We propose to use a softer trace cutting mechanism by adding a fixed tolerance parameter $\kappa$ and taking the expectation of trace coefficients under $\pi$:

$$\lambda_i = \lambda \mathbb{E}_{a \sim \pi(a|x_t)} \left[ \mathbb{I}_{[Q(x_t, a_t; \theta) \geq Q(x_t, a; \theta) - \kappa |Q(x_i, a; \theta)|]} \right] \quad (4)$$

Finally, we replace all occurrences of the max operator in Equation 3 with the expectation under $\pi$. The resulting return estimator, which we denote *Soft Watkins Q(λ)*, leads to more transitions being used and increased sample efficiency. Note that Watkins Q($\lambda$) is recovered when setting $\kappa = 0$ and $\pi = \mathcal{G}(Q)$.

**Ⓑ1 Loss and priority normalization.** As we learn a family of Q-functions which vary over a wide range of discount factors and intrinsic reward scales, we expect that the Q-functions will vary considerably in scale. This may cause the larger-scale Q-values to dominate learning and destabilize learning of smaller Q-values. This is a particular concern in environments with very small extrinsic reward scales. To counteract this effect we introduce a normalization scheme on the TD-errors similar to that used in Schaul et al. (2021). Specifically, we compute a running estimate of the standard deviation of TD-errors of the online network $\sigma_j^{\mathrm{running}}$ as well as a batch standard deviation $\sigma_j^{\mathrm{batch}}$, and compute $\sigma_j = \max(\sigma_j^{\mathrm{running}}, \sigma_j^{\mathrm{batch}}, \epsilon)$, where $\epsilon$ acts as small threshold to avoid amplification of noise past a specified scale, which we fix to 0.01 in all our experiments. We then divide the TD-errors by $\sigma_j$ when computing both the loss and priorities. As opposed to Schaul et al. (2021) we compute the running statistics on the learner, and we use importance sampling to correct the sampling distribution.

**B2** **Cross-mixture training.** Agent57 only trains the policy $j$ which was used to collect a given trajectory, but it is natural to ask whether data efficiency and robustness may be improved by training all policies at once. We propose a training loss $L$ according to the following weighting scheme between the behavior policy loss and the mean over all losses:

$$L = \eta L_{j_\mu} + \frac{1-\eta}{N} \sum_{j=0}^{N-1} L_j \tag{5}$$

where $L_j$ denotes the Q-learning loss for policy $j$, and $j_\mu$ denotes the index for the behavior policy selected by the meta-controller for the sampled trajectory. We find that an intermediate value for the mixture parameter of $\eta = 0.5$ tends to work well. To achieve better compute-efficiency we choose to deviate from the original UVFA architecture which fed a 1-hot encoding of the policy index to the LSTM, and instead modify the Q-value heads to output N sets of Q-values, one for each of the members in the family of policies introduced in Section 3. Therefore, in the end we output values for all combinations of actions and policies (see Figure 2). We note that in this setting, there is also less deviation in the recurrent states when learning across different mixture policies.

**C1** **Normalizer-free torso network.** Normalization layers are a common feature of ResNet architectures, and which are known to aid in training of very deep networks, but preliminary investigation revealed that several commonly used normalization layers are in fact detrimental to performance in our setting. Instead, we employ a variation of the NFNet architecture (Brock et al., 2021) for our policy torso network, which combines a variance-scaling strategy with scaled weight standardization and careful initialization to achieve state-of-the-art performance on ImageNet without the need for normalization layers. We adopt their use of stochastic depth (Huang et al., 2016) at training-time but omit the application of ordinary dropout to fully-connected layers as we did not observe any benefit from this form of regularization. Some care is required when using stochastic depth in conjunction with multi-step returns, as resampling of the stochastic depth mask at each timestep injects additional noise into the bootstrap values, resulting in a higher-variance return estimator. As such, we employ a *temporally-consistent* stochastic depth mask which remains fixed over the length of each training trajectory.

**C2** **Shared torso with combined loss.** Agent57 decomposes the combined Q-function into intrinsic and extrinsic components, $Q_e$ and $Q_i$, which are represented by separate networks. Such a decomposition prevents the gradients of the decomposed value functions from interfering with each other. This interference may occur in environments where the intrinsic reward is poorly aligned with the task objective, as defined by the extrinsic reward. However, the choice to use separate separate networks comes at an expensive computational cost, and potentially limits sample-efficiency since generic low-level features cannot be shared. To alleviate these issues, we introduce a shared torso for the two Q-functions while retaining separate heads.

While the form of the decomposition in Agent57 was chosen so as to ensure convergence to the optimal value-function $Q^\star$ in the tabular setting, this does not generally hold under function approximation. Comparing the combined and decomposed losses we observe a mismatch in the gradients due to the absence of cross-terms $Q_i(\theta)\frac{\partial Q_e(\theta)}{\partial \theta}$ and $Q_e(\theta)\frac{\partial Q_i(\theta)}{\partial \theta}$ in the decomposed loss:

$$\underbrace{\frac{\partial}{\partial \theta}\left[\frac{1}{2}(Q(\theta)-G)^2\right]}_{\text{combined loss}} \neq \underbrace{\frac{\partial}{\partial \theta}\left[\frac{1}{2}(Q_e(\theta)-G_e)^2 + \frac{1}{2}(\beta Q_i(\theta)-\beta G_i)^2\right]}_{\text{decomposed loss}} \tag{6}$$

$$\left[Q_e(\theta)+\beta Q_i(\theta)-G\right]\frac{\partial}{\partial \theta}\left[Q_e(\theta)+\beta Q_i(\theta)\right] \neq \left[Q_e(\theta)-G_e\right]\frac{\partial Q_e(\theta)}{\partial \theta} + \beta^2\left[Q_i(\theta)-G_i\right]\frac{\partial Q_i(\theta)}{\partial \theta} \tag{7}$$

Since we use a behavior policy induced by the total Q-function $Q = Q_e + \beta Q_i$ rather than the individual components, theory would suggest to use the combined loss instead. In addition, from a practical implementation perspective, this switch to the combined loss greatly simplifies the design choices involved in our proposed trust region method described in **A1**. The penalty paid for this choice is that the decomposition of the value function into extrinsic and intrinsic components no longer carries a strict semantic meaning. Nevertheless we do still retain an implicit inductive bias induced by multiplication of $Q_i$ with the intrinsic reward weight $\beta^j$.

**D** **Robustifying behavior via policy distillation.** Schaul et al. (2022) describe the effect of policy churn, whereby the greedy action of value-based RL algorithms may change frequently over

consecutive parameter updates. This can have a deleterious effect on off-policy correction methods: traces will be cut more aggressively than with a stochastic policy, and bootstrap values will change frequently which can result in a higher variance return estimator. In addition, our choice of training with temporally-consistent stochastic depth masks can be interpreted as learning an implicit ensemble of Q-functions; thus it is natural to ask whether we may see additional benefit from leveraging the policy induced by this ensemble.

We propose to train an explicit policy head $\pi_{\text{dist}}$ (see Figure 2) via policy distillation to match the $\epsilon$-greedy policy induced by the Q-function. In expectation over multiple gradient steps this should help to smooth out the policy over time, as well as over the ensemble, while being much faster to evaluate than the individual members of the ensemble. Similarly to the trust-region described in **A1**, we mask the policy distillation loss at any timestep where a KL constraint $C_{\text{KL}}$ is violated:

$$L_\pi = -\sum_{a,t} \mathcal{G}_\epsilon\big(Q(x_t, a; \theta)\big) \log \pi_{\text{dist}}(a|x_t; \theta) \quad \forall t \ s.t. \ KL\big(\pi_{\text{dist}}(a|x_t; \theta_T))||\pi_{\text{dist}}(a|x_t; \theta)\big) \leq C_{\text{KL}}$$

(8)

We use a fixed value of $\epsilon = 10^{-4}$ to act as a simple regularizer to prevent the policy logits from growing too large. We then use $\pi'_{\text{dist}} = \text{softmax}(\frac{\log \pi_{\text{dist}}}{\tau})$ as the target policy in our *Soft Watkins Q($\lambda$)* return estimator, where $\tau$ is a fixed temperature parameter. We tested values of $\tau$ in the range $[0, 1]$ and found that any choice of $\tau$ in this range yields improvement compared to not using the distilled policy, but values closer to 1 tend to exhibit greater stability, while those closer to 0 tend to learn more efficiently. We settled on an intermediate $\tau = 0.25$ to help balance between these two effects. We found that sampling from $\pi'_{\text{dist}}$ at behavior-time was less effective in some sparse reward environments, and instead opt to have the agent act according to $\mathcal{G}_\epsilon\big(\pi_{\text{dist}}\big)$.

## 5 EXPERIMENTS

Methods constituting MEME proposed in Section 4 aim at improving the data efficiency of Agent57, but such efficiency gains must not come at the cost of end performance. For this reason, we train our agent with a budget of 1B environment frames[3]. This budget allows us to validate that the asymptotic performance is maintained, i.e. the agent converges and is stable when improving data efficiency. Hyperparameters have been tuned over a subset of eight games, encompassing games with different reward density, scale and requiring credit assignment over different time horizons: *Frostbite*, *H.E.R.O.*, *Montezuma's Revenge*, *Pitfall!*, *Skiing*, *Solaris*, *Surround*, and *Tennis*. Specifically, we selected *Montezuma's Revenge*, *Pitfall!*, and *H.E.R.O.* as these are games exhibiting very sparse and large scale reward signal, requiring long-term credit assignment, while being partially observable. Additionally, *Solaris* and *Skiing* also require long-term credit assignment, but have medium scale and large negative scale rewards, respectively. Having a high degree of randomness in the observations is also a property of *Solaris* which our agents should be robust to. Finally, *Surround* and *Tennis* can be considered as fully-observable and having small-scale, moderately sparse rewards. In particular, due to the improvements in stability, we find it beneficial to use a higher samples per insert ratio (SPI) than the published values of Agent57 or R2D2. For all our experiments SPI is fixed to 6. An ablation on the effect of different SPI values can be found in Appendix L. An exhaustive description of the hyperparameters used is provided in Appendix A, and the network architecture in Appendix B. We report results averaged over six seeds per game for all experiments. We also show similar results with *sticky actions* (Machado et al., 2018) in Appendix K. Metrics for previous methods are computed using the mean final score per game reported in their respective publications: Agent57 (Badia et al., 2020), MuZero 20B (Schrittwieser et al., 2020), MuZero 200M (Schrittwieser et al., 2021), Muesli (Hessel et al., 2021). Data efficiency comparisons with Agent57 are based on the results reported by Badia et al. (2020).

### 5.1 SUMMARY OF RESULTS

In this section, we show that the proposed MEME agent is able to achieve a 200-fold reduction in the number of frames required for all games to surpass the human benchmark. The last game to surpass

---

[3]This corresponds to 250M agent-environment interactions due to the standard action repeat of 4 in the Atari benchmark.

Table 1: Number of games above human, capped mean, mean and median human normalized scores for the 57 Atari games. Similarly to Badia et al. (2020), we first compute the final score per game by averaging results from all random seeds, and then aggregate the scores of all 57 games. We sample three out of the six seeds per game without replacement and report the average as well as 95% confidence intervals over 10,000 repetitions of this process.

| Statistic | 200M frames | | | > 200M frames | | |
|---|---|---|---|---|---|---|
| | MEME | Muesli | MuZero | MEME | Agent57 | MuZero |
| Env frames | 200M | 200M | 200M | 1B | 90B | 20B |
| Number of games > human | $54_{(53,55)}$ | 52 | 49 | $57_{(57,57)}$ | 57 | 51 |
| Capped mean | $98_{(97,98)}$ | 92 | 89 | $100_{(100,100)}$ | 100 | 87 |
| Mean | $3305_{(3163,3446)}$ | 2523 | 2856 | $4087_{(3723,4445)}$ | 4766 | 4998 |
| Median | $848_{(829,895)}$ | 1077 | 1006 | $1185_{(1085,1325)}$ | 1933 | 2041 |
| 25th percentile | $282_{(269,303)}$ | 269 | 153 | $478_{(429,515)}$ | 387 | 276 |
| 5th percentile | $100_{(89,108)}$ | 15 | 28 | $119_{(119,120)}$ | 116 | 0 |

human score with MEME is *Pitfall!* at 390M frames, as compared to 78B frames required for *Skiing*, the last game for Agent57.

Figure 1 gives further details per game, about the required number of frames to reach the human baseline, and shows that hard exploration games such as *Private Eye*, *Pitfall!* and *Montezuma's Revenge* pose the hardest challenge for both agents, being among the last ones in which the human baseline is surpassed. This can be explained by the fact that in these games the agent might need a very large number of episodes before it is able to generate useful experience from which to learn. Notably, our agent is able to reduce the number of interactions required to outperform the human baseline in each of the 57 Atari games, by $62\times$ on average. In addition to the improved sample-efficiency, MEME shows competitive performance compared to the state-of-the-art methods shown in Table 1, while being trained for only 1B frames. When comparing with state-of-the-art agents such as MuZero (Schrittwieser et al., 2020) or Muesli (Hessel et al., 2021), we observe a similar pattern to that reported by Badia et al. (2020): they achieve very high scores in most games, as denoted by their high mean and median scores, but struggle to learn completely on a few of them (Agarwal et al., 2021). It is important to note that our agent achieves the human benchmark in all games, as demonstrated by its higher scores on the lower percentiles.

## 5.2 ABLATIONS

We analyze the contribution of all the components introduced in Section 4 through ablation experiments on the same subset of eight games. We compare methods based on the area under the score curve in order to capture end performance and data efficiency under a single metric[4]. The magnitude of this quantity varies across games, so we normalize it by its maximum value per game in order to obtain a score between 0 and 1, and report results on the eight ablation games in Figure 4. Appendix J includes full learning curves for all ablation experiments as well as ablations of other design choices (e.g. the number of times each sample is replayed on average, optimizer hyperparameters).

Results in Figure 4 demonstrate that all proposed methods play a role in the performance and stability of the agent. Disabling the trust region and using the target network for bootstrapping (**A1**) produces the most important performance drop among all ablations, likely due to the target network being slower at propagating information into the targets. Interestingly, we have observed that the trust region is beneficial even when using target networks for bootstrapping (c.f. Figure 24 in Appendix N), which suggests that the trust region may produce an additional stabilizing effect beyond what target networks alone can provide. Besides having a similar stabilizing effect, policy distillation (**D**) also speeds up convergence on some games and has less tendency to converge to local optima on some others. The Soft Watkins' $Q(\lambda)$ loss (**A2**) boosts data efficiency especially in games with sparse rewards and requiring long-term credit assignment, and we have empirically verified that

---

[4]We first compute scores at 10,000 equally spaced points in $[0, 1\text{B}]$ frames by applying piecewise linear interpolation to the scores logged during training. After averaging the scores at each of these points over all random seeds seeds, the trapezoidal rule is used to compute the area under the averaged curve.

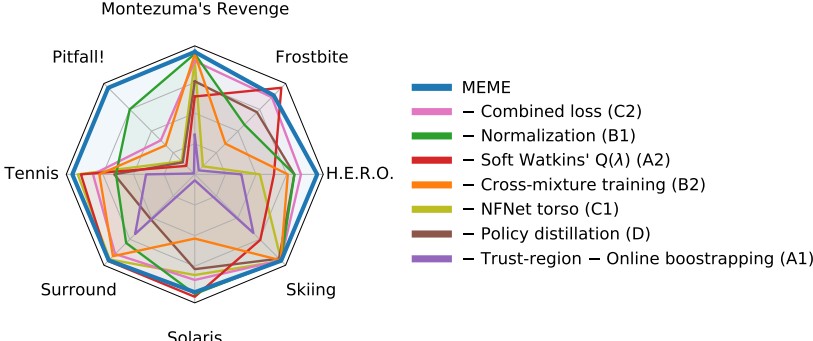

Figure 4: Results of ablating individual components. For each experiment, we first average the score over three seeds up to 1B frames, then compute the area under the score curve as it captures not only final performance but also the amount of interaction required to achieve it. As absolute values vary across games, we report relative quantities by dividing by the maximum value obtained in each game.

it uses longer traces than other losses (c.f. Figure 5 in Appendix C). Cross-mixture training (**B2**) and the combined loss (**C2**) tend to provide efficiency gains across most games. Finally, while we observe overall gains in performance from using normalization of TD errors (**B1**), the effect is less pronounced than that of other improvements. We hypothesize that the normalization has a high degree of overlap with other regularizing enhancements, such as including the trust region.

We further study the benefits of the proposed components in an agent-agnostic context, by examining their performance when using the R2D2 agent (Kapturowski et al., 2018), which we present in Appendix M.

# 6 DISCUSSION

We present a novel agent MEME that outperforms the human-level baseline in a data-efficient manner on all 57 Atari games. Our agent outperforms the human baseline across all 57 Atari games in 390M frames, using two orders of magnitude fewer interactions with the environment than Agent57, which leads to a $62\times$ speed-up on average. As Atari games are played at 60 FPS, this translates to approximately 75 days of training, compared to more than 41 years of gameplay required by Agent57.

To achieve this, the agent employs a set of improvements that address the issues apparent in the previous state-of-the art methods. Our contributions aim to achieve faster propagation of learning signals related to rare events, stabilize learning under differing value scales, improve the neural network architecture, and make updates more robust under a rapidly-changing policy. We ran ablation experiments to evaluate the contribution of each of the algorithmic and network improvements. Introducing online network bootstrapping with a trust-region has the most impact on the performance among these changes, while certain games require multiple improvements to maintain stability and performance. The increased stability enables more aggressive optimization, e.g. through higher samples per insert ratios or a centralized bandit that aggregates statistics from all actors, leading to more data-efficient learning.

Although our agent achieves above average-human performance on all 57 Atari games within 390M frames with the same set of hyperparameters, the agent is separately trained on each game. An interesting research direction to pursue would be to devise a training scheme such that the agent with the same set of weights can achieve similar performance and data efficiency as the proposed agent on all games. Furthermore, the improvements that we propose do not necessarily only apply to Agent57, and further study could analyze the impact of these components in other state-of-the-art agents. We also expect that the generality of the agent could be expanded further. While this work focuses on Atari due to its wide range of tasks and reward structures, future research is required to analyze the ability of the agent to tackle other important challenges, such as more complex observation spaces (e.g. 3D navigation, multi-modal inputs), complex action spaces, or longer-term credit assignment. All such improvements would lead MEME towards achieving greater generality.

## ACKNOWLEDGMENTS

We would like to thank Tom Schaul for his excellent feedback on improving this manuscript, and Pablo Sprechmann, Alex Vitvitskyi, Alaa Saade, Daniele Calandriello, Jake Bruce, Will Dabney, Mark Rowland, Bilal Piot, and Daniel Guo for helpful discussions throughout the development of this work.

## REPRODUCIBILITY STATEMENT

In this manuscript we made additional efforts to make sure that the explanations of the proposed methods are detailed enough to be easily reproduced by the community. Regarding the training setup, we detail the distributed setting and the computation resources used in Appendices E and G, respectively. The optimiser details are presented in Appendix D, while the comprehensive list of hyperparameters used is given in Appendix A. We explain the implementation details of the neural network architecture in Appendix B with the help of Figure 2. Finally, we give additional details about the components such as the bandit (Appendix F), the NGU and RND intrinsic rewards (Appendix H), target computation (Appendix C), and further clarify the proposed trust region with Figure 3.

## ETHICS STATEMENT

We would like to note that large scale RL experiments require a significant amount of compute, as outlined in the Appendix G. In some cases the results obtained may not justify the incurred compute and environmental costs. However, to this end, our paper greatly improves the sample-efficiency of such RL methods, and we expect its development cost and environmental impact to be amortized over many subsequent applications.

We hope that in the future, researchers will leverage the contributions presented in this paper in their work on large-scale RL, and help reduce the environmental impact their research might have.

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

## A  HYPER-PARAMETERS

Table 2: Agent Hyper-parameters.

| Parameter | Value |
|---|---|
| Num Mixtures | 16 |
| Bandit $\beta$ | 1.0 |
| Bandit $\epsilon$ | 0.5 |
| Bandit $\gamma$ | 0.999 |
| Max Discount | 0.9997 |
| Min Discount | 0.97 |
| Replay Period | 80 |
| Burn-in | 0 |
| Trace Length | 160 |
| AP Embedding Size | 32 |
| RND Scale | 0.5 |
| RND Clip Threshold | 5.0 |
| IM Reward Scale $\beta_{\text{IM}}$ | 0.1 |
| $\beta_{\text{std}}$ | 2.0 |
| Max KL $C_{\text{KL}}$ | 0.5 |
| Cross-Mixture $\eta$ | 0.5 |
| $\pi'_{\text{dist}}$ Softmax Temperature $\tau$ | 0.25 |
| Soft Watkins-Q($\lambda$) Threshold $\kappa$ | 0.01 |
| $\lambda$ | 0.95 |
| Residual Drop Rate | 0.25 |
| Eval Parameter Decay $\eta_{\text{eval}}$ | 0.995 |
| RND Stats Decay $\alpha_{\text{RND}}$ | 0.9999 |
| TD Stats Decay $\alpha_{\text{TD}}$ | 0.9999 |
| Priority Exponent | 0.6 |
| Importance Sampling Exponent | 0.4 |
| Max Priority Weight | 0.9 |
| Replay Ratio | 6.0 |
| Replay Capacity | $2 \times 10^5$ trajectories |
| Value function rescaling | $\text{sgn}(x)\left(\sqrt{x^2+1}-1\right)+0.001x$ |
| Batch Size | 64 |
| Adam $\beta_1$ | 0.9 |
| Adam $\beta_2$ | 0.999 |
| Adam $\epsilon$ | $10^{-8}$ |
| RL Adam Learning Rate | $3 \times 10^{-4}$ |
| AP Adam Learning Rate | $6 \times 10^{-4}$ |
| RND Adam Learning Rate | $6 \times 10^{-4}$ |
| RL Weight Decay | 0.05 |
| AP Weight Decay | 0.05 |
| RND Weight Decay | 0.0 |
| Gradient clipping percentile $p_{\text{clip}}$ | 0.99 |
| Gradient clipping decay $\alpha_{\text{clip}}$ | 0.999 |

Table 3: Environment Hyper-parameters.

| Parameter | Value |
|---|---|
| Input Shape | $210 \times 160$ |
| Grayscaling | True |
| Action Repeat | 4 |
| Num Stacked Frames | 1 |
| Pooled Frames | 2 |
| Max Episode Length | 108000 frames (30 minutes game time) |
| Life Loss Signal | Not used |

## B  NETWORK ARCHITECTURE

### B.1  TORSO

We use a modified version of the NFNet architecture (Brock et al., 2021). We use a simplified stem, consisting of a single $7 \times 7$ convolution with stride 4. We also forgo bottleneck residual blocks in favor of 2 layers of $3 \times 3$ convolutions, followed by a Squeeze-Excite block.

In addition, we make some minor modifications to the downsampling blocks. Specifically, we apply an activation prior to the average pooling and multiply the output by the stride in order to maintain the variance of the activations. This is then followed by a $3 \times 3$ convolution.

All convolutions use a Scaled Weight Standardization scheme (Qiao et al., 2019). The block section parameters are as follows:

- num blocks: (2, 3, 4, 4)
- num channels: (64, 128, 128, 64)
- strides: (1, 2, 2, 2)

## B.2 NON-IMAGE FEATURES

We also feed the following features into the network:

- previous action (encoded as a size-32 embedding)
- previous extrinsic reward
- previous intrinsic reward
- previous RND component of intrinsic reward
- previous Episodic component of intrinsic reward
- previous Action Prediction embedding

These are fed into a single linear layer of size 256 and activation and then concatenated with the output of the torso and input into the recurrent core.

## B.3 RECURRENT CORE

The recurrent core is composed of single LSTM with hidden size 1024. The output is then concatenated together with the core input and fed into the Action-Value and Policy heads.

## B.4 ACTION-VALUE AND POLICY HEADS

We utilize separate dueling heads for the intrinsic and extrinsic components of the value function, and a simple MLP for the policy. All heads use two hidden layers of size 1024 and output size num_actions $\times$ num_mixtures.

## C TARGET COMPUTATION AND TRACE COEFFICIENTS

As motivated in Section 4 **A2**, we introduce Soft Watkins $Q(\lambda)$ as a trade-off between aggressive trace cutting used within Retrace and Watkins $Q(\lambda)$, and the lack of off-policy correction in Peng's $Q(\lambda)$. We investigate this hypothesis by observing the average trace coefficients for each of the methods in Figure 5. The lower values of the trace coefficient $\lambda$ lead to more aggressive trace cutting as soon as the data is off-policy. Conversely, the $\lambda$ value of 1 signifies no trace cutting whatsoever. As expected, Retrace's trace coefficient is significantly lower than the other methods considered. The proposed Soft Watkins $Q(\lambda)$ has a parameter $\kappa$ that allows us to control the permissiveness of the trace-cutting which in turn affects the final trace coefficient. We ran a course sweep of this parameter over the values $\{0, .1, .01\}$ and found values of $\kappa = 0.01$ to generally perform best in our setting. As can be seen from Figure 5 this choice considerably increases the average trace coefficient of Soft Watkins' $Q(\lambda)$ relative to $\kappa = 0$, and concentrates the distribution of trace coefficients without causing it to collapse.

## D OPTIMIZER

We use separate optimizers for each of the RL, Action Prediction, and RND Networks, with the only differences between them being the choice of learning rate and weight decay. The base optimizer

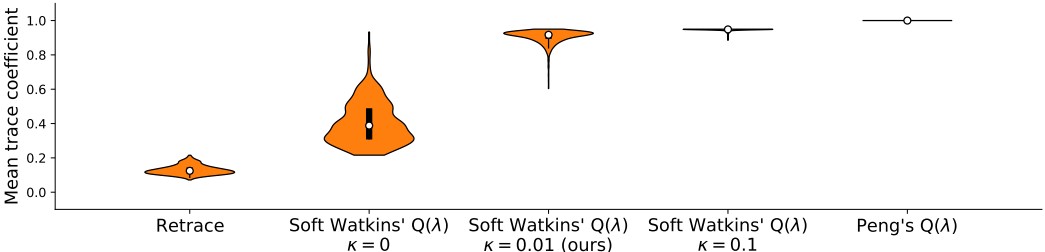

Figure 5: Average trace coefficients for each method on the set of ablation games. For each method, we average the trace length for transitions generated between 200M and 250M frames, as we observe that their values tend to stabilize after an initial transient period. Each violin plot is thus generated from `num_seeds` × `num_games` × `num_mixtures` data points.

used is AdamW with Nesterov Momentum. A linear learning rate warmup is used for the first 200 updates, to allow time for the Adam statistics to stabilize. Most notably, we employ an adaptive element-wise gradient clipping strategy whereby we maintain running estimates of the mean and standard deviation of the element-wise gradients using an Exponential Moving Average (EMA) with $\alpha_{\text{clip}} = 0.999$, and only clip when the gradient magnitude exceeds the mean by some factor $c$ of the standard deviation. Concretely, the factor $c$ is determined by specifying a percentile $p_{\text{clip}}$ (that we fix to $0.99$) which is then input to the inverse CDF of a standard normal distribution. In general we found this clipping strategy to yield a small but consistent improvement in stability across many environments compared to global norm clipping. We ablate some of the optimizer choices in Figure 19.

## E    DISTRIBUTED SETTING

All experiments are run using a distributed setting. The experiment consists of the actors, learner, bandit and evaluators, as well as a replay buffer.

The actors and evaluators are the two types of workers that draw samples from the environment. Since actors collect experience with non-greedy policies, we follow the common practice in this type of agent and report scores from separate evaluator processes that continually execute the greedy policy and whose experience is not added to the replay buffer (Kapturowski et al., 2018). Therefore, only the actor workers write to the replay buffer, while the evaluation workers are used purely for reporting the performance. The evaluation scheme differs from R2D2 (Kapturowski et al., 2018) in that a separate set of eval parameters are maintained, which are computed as an EMA of the online network with $\eta_{\text{eval}} = 0.995$; and these eval parameters are continually updated throughout each episode. We observed that the use of these eval parameters provided a consistent performance boost across almost all environments, but we continue to use the online network for the actors in order to obtain more varied experience.

In the replay buffer, we store fixed-length sequences of $(r_t^e, r_t^{\text{NGU}}, x_t, a_t)$ tuples. These sequences never cross episode boundaries. For Atari, we apply the standard DQN pre-processing, as used in R2D2. The replay buffer is split into 8 shards, to improve robustness due to computational constraints, with each shard maintaining an independent prioritisation of the entries. We use prioritized experience replay with the same prioritization scheme proposed by Kapturowski et al. (2018) which used a weighted mixture of max and mean TD-errors over the sequence. Each of the actor workers writes to a specific shard which is consistent throughout training. The replay buffer is implemented using Reverb (Cassirer et al., 2021).

Given a single batch of trajectories we unroll both online and target networks on the same sequence of states to generate value estimates. These estimates are used to execute the learner update step, which updates the model weights used by the actors, and the exponential moving average (EMA) of the weights used by the evaluator models, as this yields best performance which we report.

Acting in the environment is accelerated by sending observations from actors and evaluators to a shared server that runs batched inference. The remote inference server allows multiple clients such as

actor and evaluator workers to connect to it, and executes their inputs as a batch on the corresponding inference models. The actor and evaluator inference model parameters are queried periodically from the learner. Also, the recurrent state is persisted on the inference server so that the actor does need to communicate it. However, the episodic memory lookup required to compute the intrinsic reward is performed locally on actors to reduce the communication overhead.

At the beginning of each episode, parameters $\beta$ and $\gamma$ are queried from the bandit worker, i.e. meta-controller. The parameters are selected from a set of coefficients $\{(\beta_j, \gamma_j)\}_{j=0}^{N-1}$ with $N = 16$, which correspond to the $N$-heads of the network. The actors query optimal $(\beta, \gamma)$ tuples, while the evaluators query the tuple corresponding to the greedy action. After each actor episode, the bandit stats are updated based on the episode rewards by updating the distribution over actions, according to Discounted UCB (Garivier & Moulines, 2011).

The following subsections describe how actors, evaluators, and learner are run in each stage.

### LEARNER

- Sample a sequence of extrinsic rewards $r_t^e$, intrinsic rewards $r_t^{\text{NGU}}$, observations $x_t$ and actions $a_t$, from the replay buffer.
- Use Q-network to learn from $(r_t^e, r_t^{\text{NGU}}, x, a)$ with our modified version of Watkins' Q($\lambda$) (Watkins & Dayan, 1992) using the same procedure as in R2D2.
- Compute the actor model weights and EMA for the evaluator model weights.
- Use the sampled sequences to train the action prediction network in NGU.
- Use the sampled sequences to train the predictor of RND.

### ACTOR

- Query optimal bandit action $(\beta, \gamma)$.
- Obtain $x_t$ from the environment.
- Obtain $r_t^{\text{NGU}}$ and $a_t$ from the inference model.
- Insert $x_t$, $a_t$, $r_t^{\text{NGU}}$ and $r_t^e$ in the replay buffer.
- Step on the environment with $a_t$.

### EVALUATOR

- Query greedy bandit action $(\beta, \gamma)$.
- Obtain $x_t$ from the environment.
- Obtain $r_t^{\text{NGU}}$ and $a_t$ from the inference model.
- Step on the environment with $a_t$.

### BANDIT

- Periodically checkpoints bandit action values.
- Queried for optimal action by actors.
- Queried for greedy action by evaluators.
- Updates the stats when actors pass the episode rewards for a certain action.

## F  BANDIT IMPLEMENTATION

While Agent57 maintains a separate bandit for each actor, we instead utilize a centralized bandit worker. The bandit selects between a family of policies generated by tuples of intrinsic reward weight and discount factor $(\beta_i, \gamma_i)$, parameterized as:

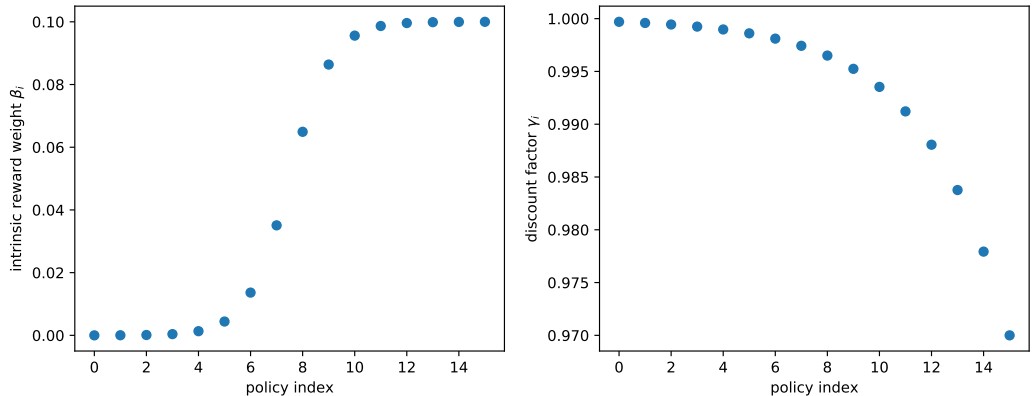

Figure 6: $\beta$ and $\gamma$ for each of the 16 policies.

$$\beta_i = \begin{cases} 0 & \text{if } i = 0 \\ \beta_{\text{IM}} & \text{if } i = N - 1 \\ \beta_{\text{IM}}\sigma(8\frac{2i-(N-2)}{N-2}) & \text{otherwise} \end{cases} \tag{9}$$

$$\gamma_i = 1 - \exp\left(\frac{N-1-i}{N-1}\log(1-\gamma_{\text{max}}) + \frac{i}{N-1}\log(1-\gamma_{\text{max}})\right) \tag{10}$$

where $N$ is the number of policies, $i$ is the policy index, $\sigma$ is the sigmoid function, $\beta_{\text{IM}}$ is the maximum intrinsic reward weight, and $\gamma_{\text{max}}$ and $\gamma_{\text{int}}$ are the maximum and minimum discount factors, respectively.

At the beginning of each episode an actor will sample a policy index with which to act for the duration of the episode. At the end of which, the actor will update the bandit with the obtained extrinsic return for that policy. We use a discounted variant of UCB-Tuned bandit algorithm (Garivier & Moulines (2008) and Auer et al. (2002)). In practice the bandit hyper-parameters did not seem to be very important. We hypothesize that the use of cross-mixture training may reduce the sensitivity of the agent to these parameters, though we have not explored this relationship thoroughly.

## G COMPUTE RESOURCES

For the experiments we used the TPUv4, with the $2 \times 2 \times 1$ topology used for the learner. Acting is accelerated by sending observations from actors to a shared server that runs batched inference using a $1 \times 1 \times 1$ TPUv4, which is used for inference within the actor and evaluation workers.

On average, the learner performs 3.8 updates per second. The rate at which environment frames are written to the replay buffer by the actors is approximately 12,970 frames per second.

Each experiment consists of 64 actors with 2 threads, each of them acting with their own independent instance of the environment. The collected experience is stored in the replay buffer split in 8 shards, each with independent prioritization. This accumulated experience is used by a single learner worker, while the performance is evaluated on 5 evaluator workers.

## H INTRINSIC REWARDS

### H.1 RANDOM NETWORK DISTILLATION

The Random Network Distillation (RND) intrinsic reward (Burda et al., 2019) is computed by introducing a random, untrained convolutional network $g : \mathcal{X} \to \mathbb{R}^d$, and training a network $\hat{g} : \mathcal{X} \to \mathbb{R}^d$ to predict the outputs of $g$ on all the observations that are seen during training by minimizing the prediction error $\text{err}_{\text{RND}}(x_t) = ||\hat{g}(x_t; \theta) - g(x_t)||^2$ with respect to $\theta$. The intuition is

that the prediction error will be large on states that have been visited less frequently by the agent. The dimensionality of the random embedding, $d$, is a hyperparameter of the algorithm.

The RND intrinsic reward is obtained by normalising the prediction error. In this work, we use a slightly different normalization from that reported in Burda et al. (2019). The RND reward at time $t$ is given by

$$r_t^{\text{RND}} = \frac{\text{err}_{\text{RND}}(x_t)}{\sigma_e} \tag{11}$$

where $\sigma_e$ is the running standard deviation of $\text{err}_{\text{RND}}(x_t)$. As with the TD-error statistics, we compute $\sigma_e$ on the learner using importance sampling weights to correct the sampling distribution.

## H.2 NEVER GIVE UP

The NGU intrinsic reward modulates an episodic intrinsic reward, $r_t^{\text{episodic}}$, with a life long signal $\alpha_t$:

$$r_t^{\text{NGU}} = r_t^{\text{episodic}} \cdot \min\{\max\{\alpha_t, 1\}, L\}, \tag{12}$$

where $L$ is a fixed maximum reward scaling. The life-long novelty signal is computed using RND with the normalisation:

$$\alpha_t = \frac{\text{err}_{\text{RND}}(x_t) - \mu_e}{\sigma_e} \tag{13}$$

where $\text{err}_{\text{RND}}(x_t)$ is the prediction error described in Appendix H.1, and $\mu_e$ and $\sigma_e$ are its running mean and standard deviation, respectively. The episodic intrinsic reward at time $t$ is computed according to formula:

$$r_t^{\text{episodic}} = \frac{1}{\sqrt{\sum_{f(x_i) \in N_k} K(f(x_t), f(x_i))} + c} \tag{14}$$

where $N_k$ is the set containing the $k$-nearest neighbors of $f(x_t)$ in $M$, $c$ is a constant and $K : \mathbb{R}^p \times \mathbb{R}^p \to \mathbb{R}^+$ is a kernel function satisfying $K(x, x) = 1$ (which can be thought of as approximating pseudo-counts Badia et al. (2019)). Algorithm 1 shows a detailed description of how the episodic intrinsic reward is computed. Below we describe the different components used in Algorithm 1:

- $M$: episodic memory containing at time $t$ the previous embeddings $\{f(x_0), f(x_1), \ldots, f(x_{t-1})\}$. This memory starts empty at each episode
- $k$: number of nearest neighbours
- $N_k = \{f(x_i)\}_{i=1}^k$: set of $k$-nearest neighbours of $f(x_t)$ in the memory $M$; we call $N_k[i] = f(x_i) \in N_k$ for ease of notation
- $K$: kernel defined as $K(x, y) = \frac{\epsilon}{\frac{d^2(x,y)}{d_m^2} + \epsilon}$ where $\epsilon$ is a small constant, $d$ is the Euclidean distance and $d_m^2$ is a running average of the squared Euclidean distance of the $k$-nearest neighbors [5]
- $c$: pseudo-counts constant
- $\xi$: cluster distance
- $s_m$: maximum similarity
- $f(x)$: action prediction network output for observation $x$ as in Badia et al. (2020).

# I OTHER THINGS WE TRIED

## I.1 FUNCTIONAL REGULARIZATION IN PLACE OF TRUST REGION

Concurrently with the development of our trust region method we experimented with using an explicit L2 regularization term in the loss, acting on the difference between the online and target networks, similar to (Piche et al., 2021). Prior to implementation of our normalization scheme we found that this method stabilized learning early on in training but was prone to eventually becoming unstable if run for long enough. With normalization this instability did not occur, but sample efficiency was worse compared to the trust region in most instances we observed.

---

[5] As opposed to Agent57 which stored $d_m^2$ separately per actor, we aggregate over all actors

---

**Algorithm 1:** Computation of the episodic intrinsic reward at time $t$: $r_t^{\text{episodic}}$.

---

**Input** : $M$; $k$; $f(x_t)$; $c$; $\epsilon$; $\xi$; $x_m$; $d_m^2$

**Output:** $r_t^{\text{episodic}}$

Compute the $k$-nearest neighbours of $f(x_t)$ in $M$ and store them in a list $N_k$

Create a list of floats $d_k$ of size $k$

```
/* The list d_k will contain the distances between the embedding f(x_t) and its neighbours
   N_k.                                                                                    */
```

**for** $i \in \{1, \ldots, k\}$ **do**

$\quad\mid\quad d_k[i] \leftarrow d^2(f(x_t), N_k[i])$

**end**

Update the moving average $d_m^2$ with the list of distances $d_k$

```
/* Normalize the distances d_k with the updated moving average d_m^2.                      */
```

$d_n \leftarrow \frac{d_k}{d_m^2}$

```
/* Cluster the normalized distances d_n i.e.  they become 0 if too small and 0_k is a list of
   k zeros.                                                                                 */
```

$d_n \leftarrow \max(d_n - \xi, 0_k)$

```
/* Compute the Kernel values between the embedding f(x_t) and its neighbours N_k.           */
```

$K_v \leftarrow \frac{\epsilon}{d_n + \epsilon}$

```
/* Compute the similarity between the embedding f(s_t) and its neighbours N_k.              */
```

$s \leftarrow \sqrt{\sum_{i=1}^{k} K_v[i]} + c$

```
/* Compute the episodic intrinsic reward at time t:  r_t^i.                                 */
```

**if** $x > x_m$ **then**

$\quad\mid\quad r_t^{\text{episodic}} \leftarrow 0$

**else**

$\quad\mid\quad r_t^{\text{episodic}} \leftarrow 1/s$

---

## I.2 Approximate Thompson Sampling

We considered using an approximate Thompson Sampling scheme scheme similar to Bootstrapped DQN (Osband et al., 2016) whereby the stochastic depth mask was fixed for some period of time at inference (such as once per episode, or every 100 timesteps). We observed some marginal benefit in certain games, but in our view this difference was not enough to justify the added complexity of implementation. We hypothesize that the added exploration this provides is not significant when a strong intrinsic reward is already present in the learning algorithm, but it may have a larger effect if this is not the case.

## I.3 Mixture of Online and Replay Data

We considered using a mixture of online and replay data as was done in the Muesli agent (Hessel et al., 2021). This was beneficial for overall stability, but it also degraded performance in the harder exploration games such as *Pitfall!*. We were not able to find an easy remedy for this so we did not investigate further in this direction.

## I.4 Estimating $Q_e(\theta) + \beta Q_i(\theta)$

Agent57 uses two neural networks with completely independent weights to estimate $Q_e$ and $Q_i$. As mentioned in the work, this provides the network with more robustness to the different scales and variance that $r_e$ and $r_i$ have for many tasks.

MEME changes the separation of networks, whereby the $Q_e$ and $Q_i$ are still estimated separately, but they share a common torso and recurrent core. However, since many of the components we introduce are geared toward improving stability, even this separation may no longer be necessary. To analyze this we run an experiment where the agent network has a single head that estimates $Q_e(\theta) + \beta Q_i(\theta)$. Note that in this case we still estimate $N$ sets of Q-values. Indeed, as results of Figure 7 show, we observe similar results as our proposed method. This indicates that the inductive bias that was introduced in maintaining separate heads for intrinsic and extrinsic $Q$-values is no longer important.

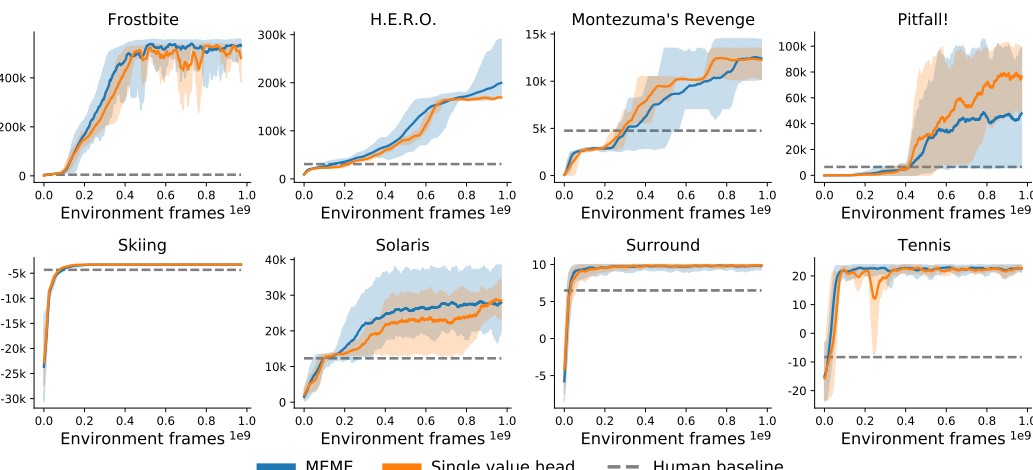

Figure 7: Results for the two approaches to estimating the total loss.

# J ADDITIONAL RESULTS

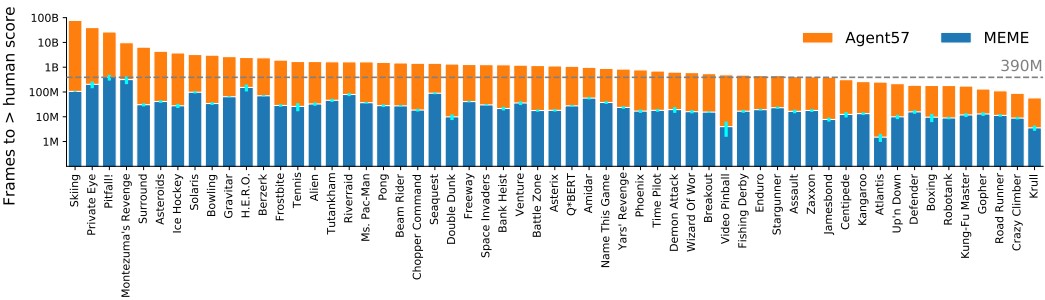

Figure 8: Number of environment frames required by agents to outperform the human baseline on each game (in log-scale). Lower is better. Error bars represent the standard deviation over seeds for each game. On average, MEME achieves above human scores using $62\times$ fewer environment interactions than Agent57. The smallest improvement is $10\times$ (*Road Runner*), the maximum is $734\times$ (*Skiing*), and the median across the suite is $36\times$.

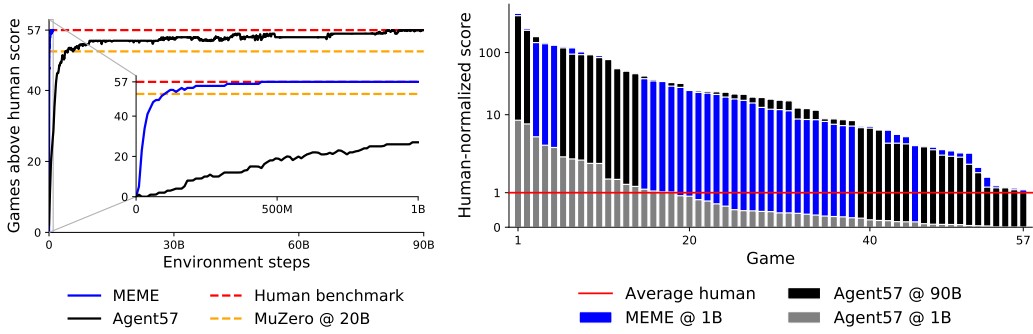

Figure 9: Comparison with Agent57. *Left:* number of games with scores above the human benchmark. *Right:* human-normalized scores per game at different interaction budgets, sorted from highest to lowest. Our agent outperforms the human benchmark in 390M frames, two orders of magnitude faster than Agent57, and achieves similar end scores while reducing the training budget from 90B to 1B frames.

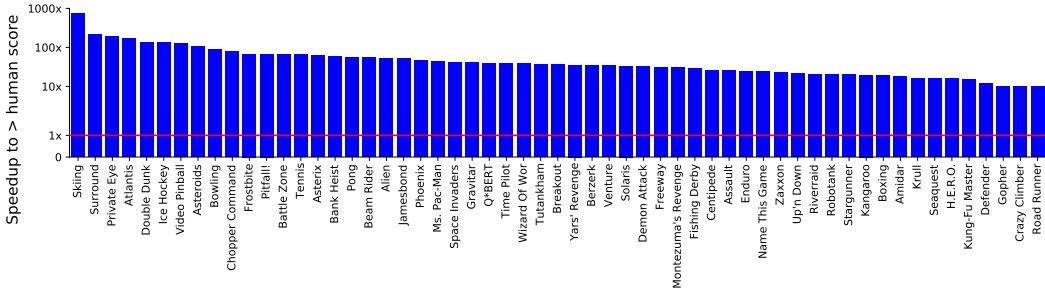

Figure 10: Speedup in reaching above human performance for the first time, computed as the ratio between the orange and blue bars in Figure 1.

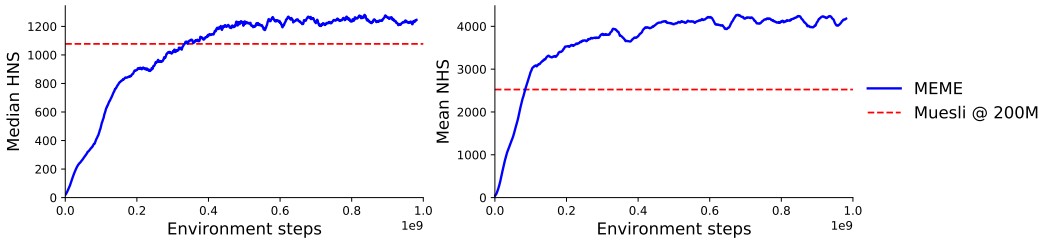

Figure 11: Median and mean scores over the course of training.

## K  STICKY ACTIONS RESULTS

This section reproduces the main results of the paper, but enabling *sticky actions* (Machado et al., 2018) during both training and evaluation. Since our agent does not exploit the determinism in the original Atari benchmark, it is still able to outperform the human baseline with *sticky actions* enabled. We observe a slight decrease in mean scores, which we attribute to label corruption in the action prediction loss used to learn the controllable representations used in the episodic reward computation: due to the implementation of sticky actions proposed by Machado et al. (2018) the agent actions are ignored in a fraction of the timesteps. This phenomenon is aggravated by the frame stacking used in the standard Atari pre-processing, as the action being executed in the environment can vary within each stack of frames. We hypothesize that the gap between the two versions of the environment would be much smaller with a different implementation of *sticky actions* that did not corrupt the action labels used by the representation learning module.

All reported results are the average over six different random seeds.

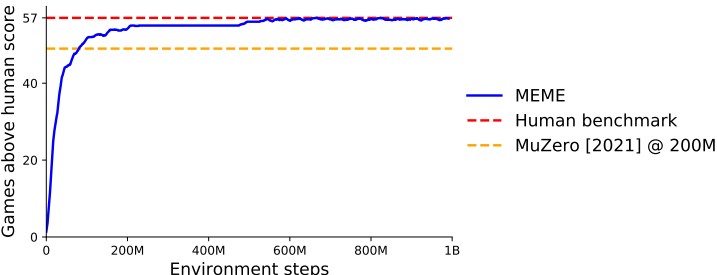

Figure 12: Number of games with scores above the human benchmark when training and evaluating with *sticky actions* (Machado et al., 2018).

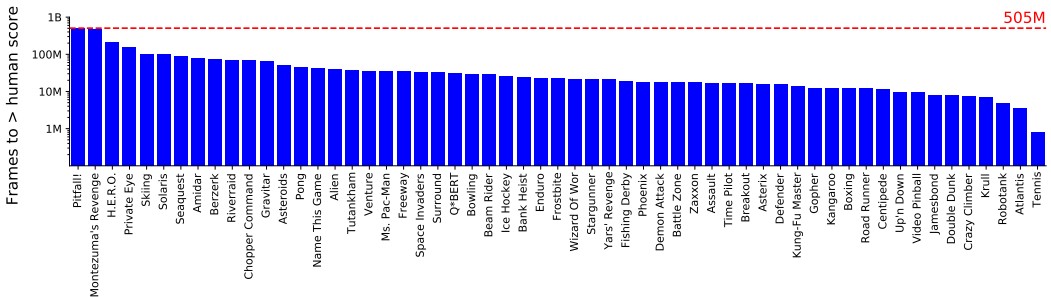

Figure 13: Number of environment frames required by agents to outperform the human baseline on each game when training and evaluating with *sticky actions* (Machado et al., 2018). The human baseline is outperformed after 505M frames.

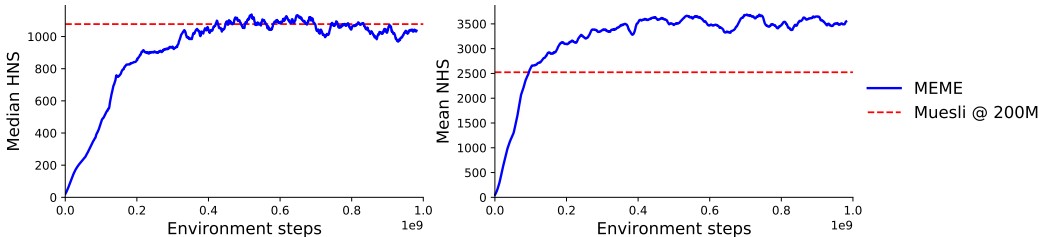

Figure 14: Median and mean scores over the course of training when training and evaluating with *sticky actions* (Machado et al., 2018).

Table 4: Number of games above human, capped mean, mean and median human normalized scores for the 57 Atari games when training and evaluating with *sticky actions* (Machado et al., 2018). Metrics for previous methods are computed using the final score per game reported in their respective publications: MuZero (Schrittwieser et al., 2021), Muesli (Hessel et al., 2021).

| Statistic | 200M frames | | | > 200M frames |
|---|---|---|---|---|
| | **MEME** | **Muesli** | **MuZero** | **MEME** |
| Env frames | 200M | 200M | 200M | 1B |
| Number of games > human | **54** | 52 | 49 | **57** |
| Capped mean | **97.51** | 92.52 | 89.78 | **100.0** |
| Mean | **2967.52** | 2523.99 | 2856.24 | **3462.93** |
| Median | 830.57 | **1077.47** | 1006.4 | **1074.25** |
| 25th percentile | **299.65** | 269.25 | 153.1 | **402.56** |
| 5th percentile | **103.86** | 15.91 | 28.76 | **118.78** |

## L   EFFECT OF SAMPLES PER INSERT RATIO

Results of the ablation on the amount of replay that the learner performs per sequence of experience that the actors produce can be seen in Figure 15. We can see that, while a samples per insert ratio (SPI) of 10 still provides moderate boosts in data efficiency in games such as *H.E.R.O.*, *Montezuma's Revenge* and *Pitfall!*, it is not as pronounced as the increase that is seen from SPI of 3 to 6. This implies that with an SPI of 10 we obtain a much worse return in terms of wall-clock time as we replay more frequently.

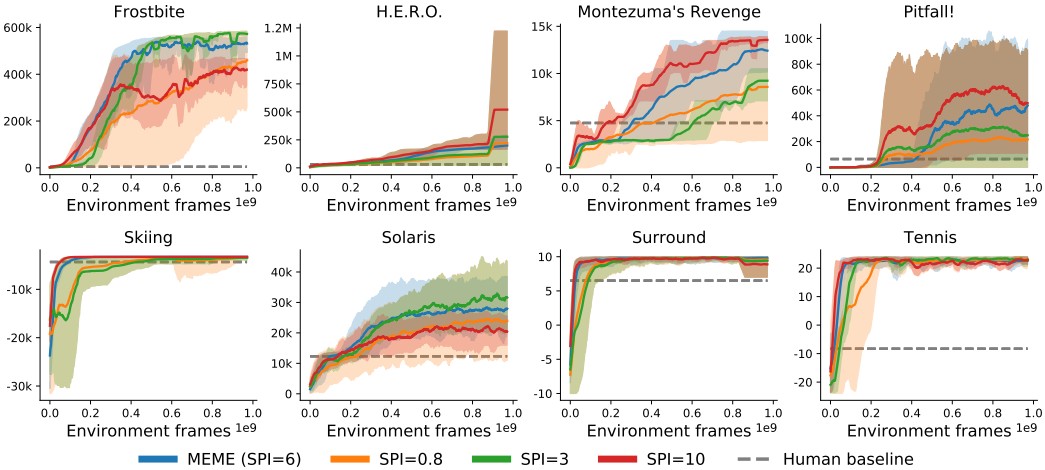

Figure 15: Sweep over sample per insert ratios on the full ablation set.

## M    R2D2 ABLATIONS

We further study the benefits of the proposed components in an agent-agnostic context, by examining their performance when used with the R2D2 agent (Kapturowski et al., 2018), as shown in Figure 16. Similarly to the original ablations of MEME, we can see that the trust region and online bootstrapping constitute the most important component, which once removed deteriorates the performance significantly. In general, the rest of the components are beneficial as well, as MEME R2D2 can be seen to obtain human-level performance on the largest number of games; although the set of Atari games used in this ablation is not fully representative as *Montezuma's Revenge*, *Pitfall!* and *Skiing* are not expected to be solvable without intrinsic motivation or the ability to leverage larger discount factors.

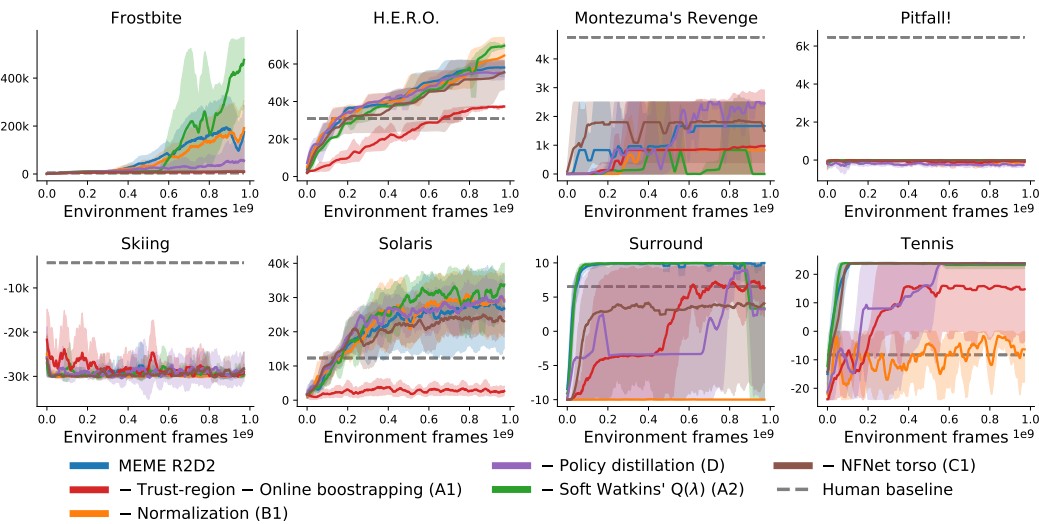

Figure 16: Ablation study of the proposed components while using the R2D2 agent.

# N    FULL ABLATIONS

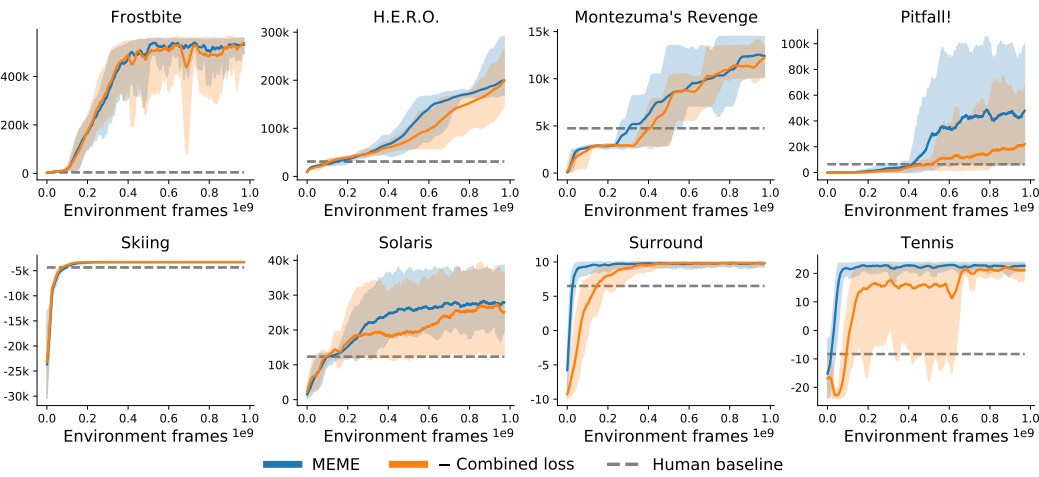

Figure 17: Comparison between our combined loss and the separate losses used by Agent57 on the full ablation set.

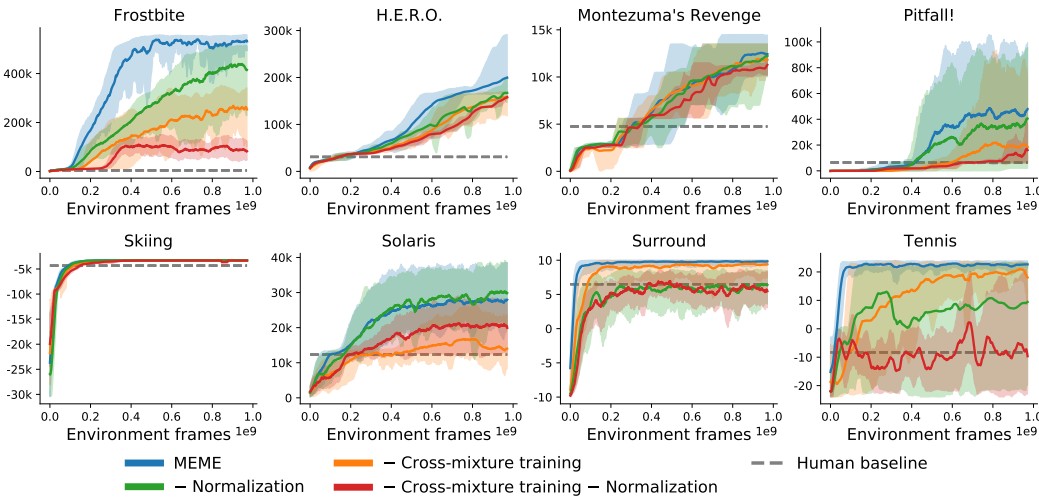

Figure 18: Results without cross-mixture training and TD normalization on the full ablation set.

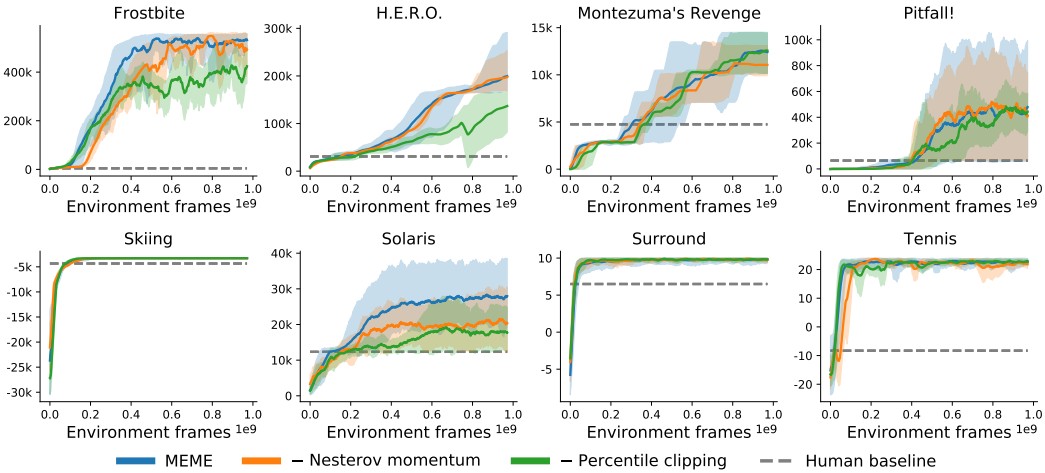

Figure 19: Ablation experiments over different optimizer features.

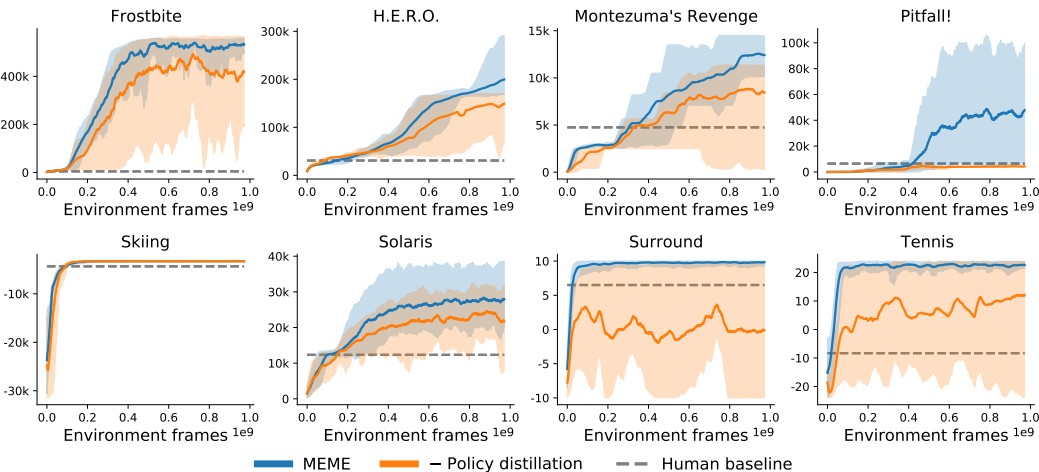

Figure 20: Results without policy distillation on the full ablation set.

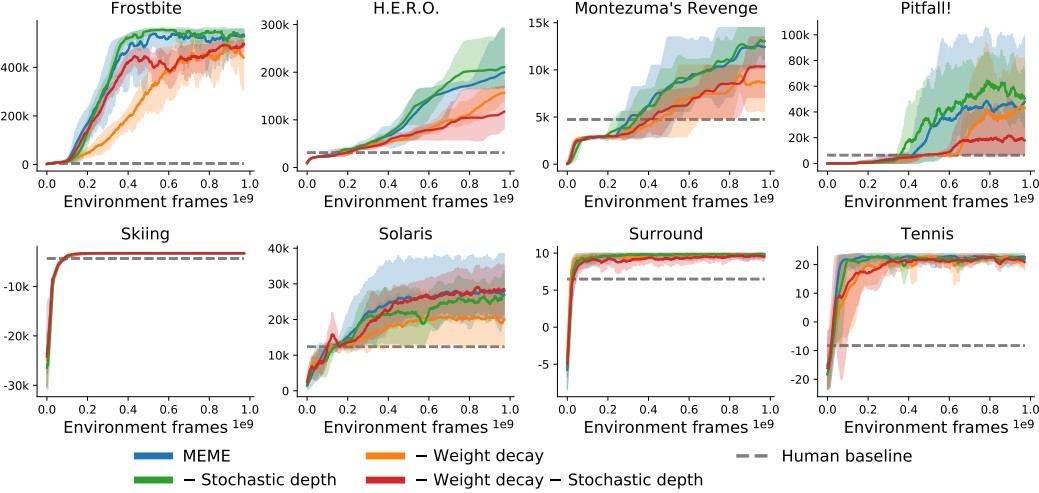

Figure 21: Results for agents with different amounts of regularization on the full ablation set.

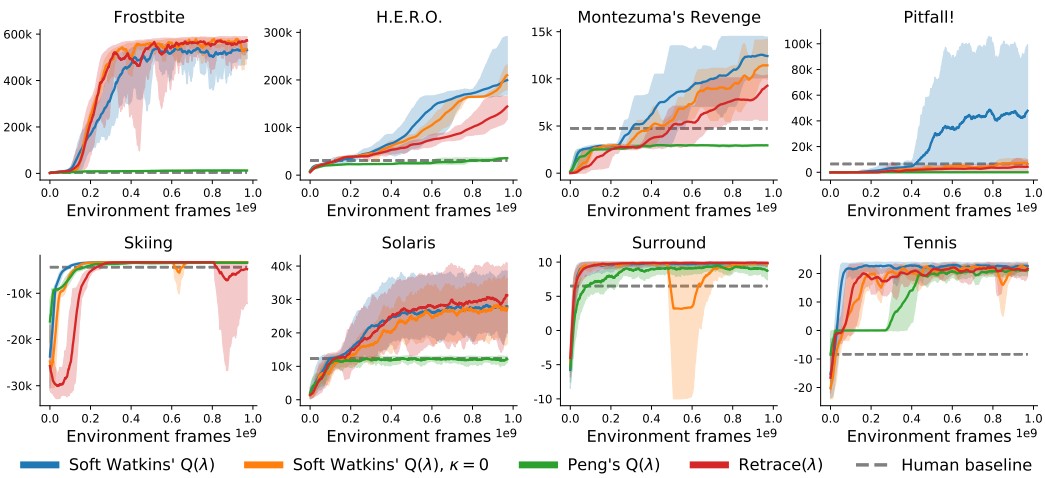

Figure 22: Results with different learning targets on the full ablation set.

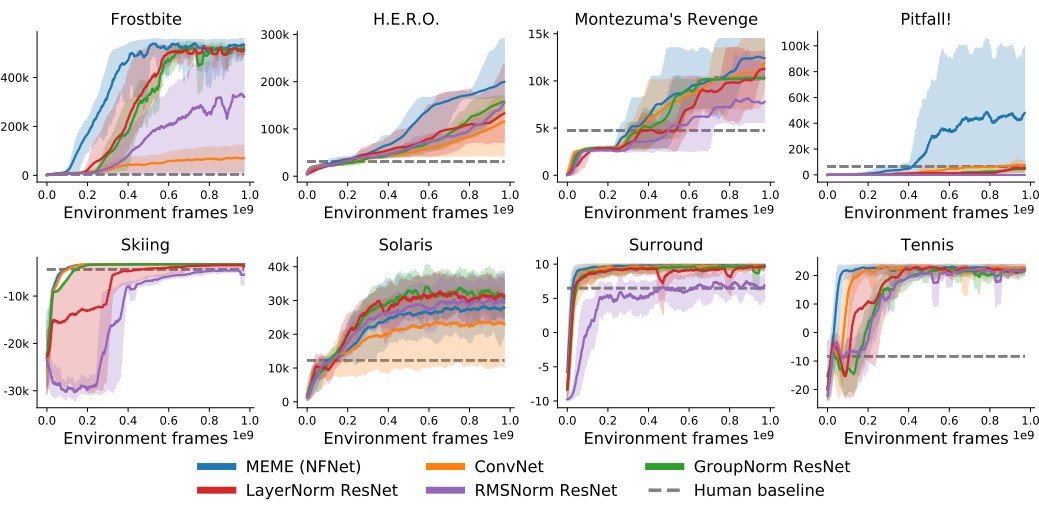

Figure 23: Results for agents with different torsos on the full ablation set.

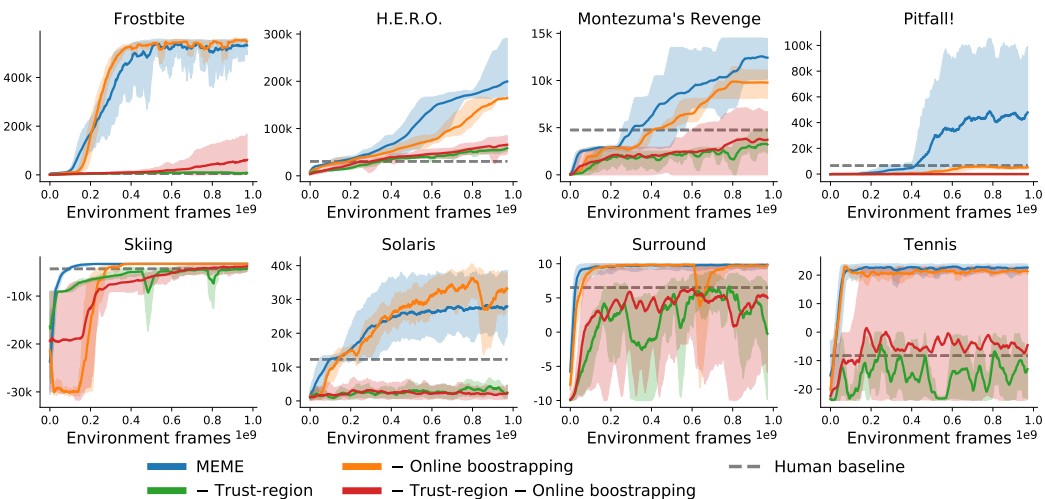

Figure 24: Results for agents without online bootstrapping and trust-region on the full ablation set.

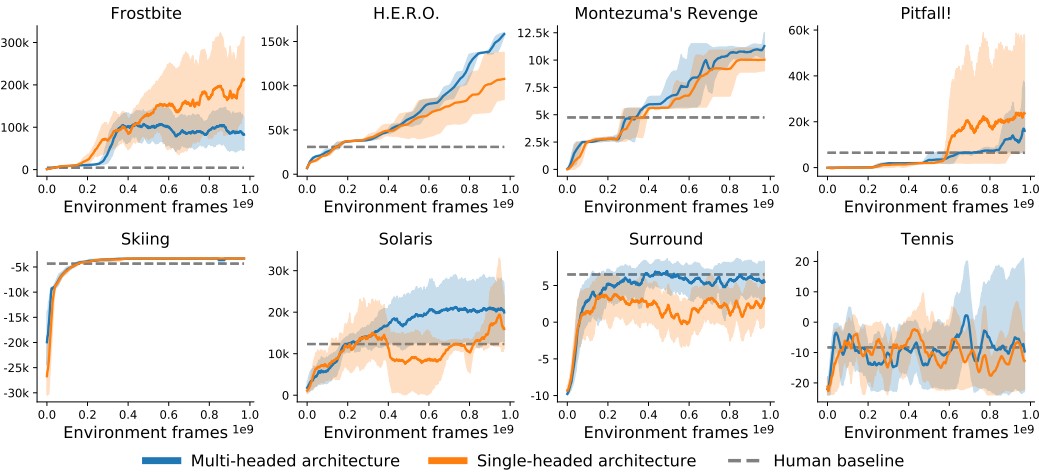

Figure 25: Comparison of head architectures. The single-headed architecture is similar to the one in Agent57, where the network is conditioned on a one-hot encoding of the mixture id. All experiments are run without cross-mixture training and TD normalization for fairness. These results demonstrate that our multi-headed architecture, introduced to enable efficient computation of Q values for all mixtures in parallel, does not degrade performance.

# O    SCORES PER GAME

Table 5: Scores per game.

| Game | Agent57 @ 1B | Agent57 @ 90B | MEME @ 200M | MEME @ 1B |
|---|---|---|---|---|
| Alien | $3094.46 \pm 1682.49$ | $\mathbf{252889.92 \pm 31085.48}$ | $41048.78 \pm 8702.03$ | $83683.43 \pm 16688.58$ |
| Amidar | $329.99 \pm 124.14$ | $\mathbf{28671.19 \pm 1046.52}$ | $7363.47 \pm 1033.44$ | $14368.90 \pm 2775.86$ |
| Assault | $1183.25 \pm 482.14$ | $\mathbf{49198.37 \pm 9469.92}$ | $33266.67 \pm 6143.71$ | $46635.86 \pm 14846.53$ |
| Asterix | $2777.67 \pm 697.39$ | $763476.87 \pm 105395.98$ | $\mathbf{861780.67 \pm 97588.31}$ | $769803.92 \pm 143061.91$ |
| Asteroids | $2422.64 \pm 412.96$ | $105058.07 \pm 45380.23$ | $217586.98 \pm 17304.43$ | $\mathbf{364492.07 \pm 13982.31}$ |
| Atlantis | $44844.67 \pm 15838.85$ | $1508119.97 \pm 24913.10$ | $1535634.17 \pm 20700.45$ | $\mathbf{1669226.33 \pm 3906.17}$ |
| Bank Heist | $364.26 \pm 156.87$ | $17274.04 \pm 12145.09$ | $15563.35 \pm 14565.74$ | $\mathbf{87792.55 \pm 104611.67}$ |
| Battle Zone | $19065.86 \pm 999.82$ | $\mathbf{868540.57 \pm 26523.78}$ | $733206.67 \pm 84295.20$ | $776770.00 \pm 19734.15$ |
| Beam Rider | $2056.68 \pm 131.88$ | $\mathbf{283845.01 \pm 11289.64}$ | $68534.71 \pm 4443.34$ | $51870.20 \pm 2906.10$ |
| Berzerk | $614.26 \pm 84.32$ | $31565.12 \pm 34278.20$ | $7003.12 \pm 2592.61$ | $\mathbf{38838.35 \pm 14783.99}$ |
| Bowling | $34.00 \pm 4.19$ | $240.33 \pm 18.96$ | $\mathbf{261.83 \pm 2.30}$ | $261.74 \pm 8.42$ |
| Boxing | $97.92 \pm 1.08$ | $\mathbf{99.93 \pm 0.03}$ | $99.77 \pm 0.17$ | $99.85 \pm 0.14$ |
| Breakout | $62.09 \pm 34.67$ | $696.94 \pm 63.20$ | $747.62 \pm 64.06$ | $\mathbf{831.08 \pm 6.18}$ |
| Centipede | $14411.24 \pm 1780.60$ | $\mathbf{348288.74 \pm 11957.45}$ | $112609.74 \pm 42701.73$ | $245892.18 \pm 39060.78$ |
| Chopper Command | $3535.40 \pm 1031.08$ | $\mathbf{959747.29 \pm 10225.76}$ | $842327.17 \pm 168089.18$ | $912225.00 \pm 112906.65$ |
| Crazy Climber | $85166.13 \pm 9121.39$ | $\mathbf{456653.80 \pm 13192.11}$ | $295413.67 \pm 5974.67$ | $339274.67 \pm 14818.41$ |
| Defender | $33613.13 \pm 6856.21$ | $\mathbf{666433.14 \pm 17493.30}$ | $518605.50 \pm 18011.92$ | $543979.50 \pm 7639.27$ |
| Demon Attack | $1786.19 \pm 911.24$ | $140474.21 \pm 2931.07$ | $139349.75 \pm 1927.91$ | $\mathbf{142176.58 \pm 1223.59}$ |
| Double Dunk | $-21.76 \pm 0.52$ | $23.64 \pm 0.06$ | $23.60 \pm 0.06$ | $\mathbf{23.70 \pm 0.44}$ |
| Enduro | $437.34 \pm 97.19$ | $2349.03 \pm 7.46$ | $2338.62 \pm 38.96$ | $\mathbf{2360.64 \pm 3.19}$ |
| Fishing Derby | $-43.36 \pm 19.65$ | $\mathbf{83.42 \pm 4.11}$ | $67.19 \pm 5.30$ | $77.05 \pm 3.97$ |
| Freeway | $22.46 \pm 0.49$ | $32.13 \pm 0.71$ | $33.82 \pm 0.14$ | $\mathbf{33.97 \pm 0.02}$ |
| Frostbite | $980.15 \pm 581.61$ | $507775.65 \pm 35925.95$ | $136691.77 \pm 35672.33$ | $\mathbf{526239.50 \pm 18289.50}$ |
| Gopher | $9760.20 \pm 1790.84$ | $98786.14 \pm 4600.57$ | $117557.53 \pm 3264.72$ | $\mathbf{119457.53 \pm 4077.33}$ |
| Gravitar | $666.43 \pm 304.08$ | $18180.26 \pm 627.48$ | $13049.67 \pm 272.66$ | $\mathbf{20875.00 \pm 844.41}$ |
| H.E.R.O. | $8850.63 \pm 1313.70$ | $102145.96 \pm 50561.27$ | $33872.29 \pm 6917.14$ | $\mathbf{199880.60 \pm 44074.56}$ |
| Ice Hockey | $-14.87 \pm 1.71$ | $\mathbf{62.33 \pm 5.77}$ | $26.07 \pm 4.48$ | $47.22 \pm 4.41$ |
| Jamesbond | $534.53 \pm 408.19$ | $107266.11 \pm 15584.58$ | $\mathbf{137333.17 \pm 21939.77}$ | $117009.92 \pm 55411.15$ |
| Kangaroo | $3178.86 \pm 996.26$ | $\mathbf{18505.37 \pm 5016.41}$ | $15863.50 \pm 675.45$ | $17311.17 \pm 419.17$ |
| Krull | $9179.71 \pm 1222.12$ | $\mathbf{194179.21 \pm 21451.96}$ | $157943.83 \pm 26699.67$ | $155915.32 \pm 43127.45$ |
| Kung-Fu Master | $31613.73 \pm 8854.35$ | $192616.60 \pm 9019.89$ | $364755.65 \pm 387274.47$ | $\mathbf{476539.53 \pm 518479.85}$ |
| Montezuma's Revenge | $200.43 \pm 192.50$ | $8666.10 \pm 2928.92$ | $2863.00 \pm 117.94$ | $\mathbf{12437.00 \pm 1648.44}$ |
| Ms. Pac-Man | $3348.12 \pm 630.07$ | $\mathbf{57402.56 \pm 3077.27}$ | $22853.12 \pm 2843.55$ | $29747.91 \pm 2472.33$ |
| Name This Game | $4463.97 \pm 1747.50$ | $\mathbf{48644.27 \pm 2390.83}$ | $31369.93 \pm 2637.53$ | $40077.73 \pm 2274.25$ |
| Phoenix | $7359.04 \pm 5542.07$ | $\mathbf{858909.13 \pm 37669.01}$ | $602393.53 \pm 43967.79$ | $849969.25 \pm 43573.52$ |
| Pitfall! | $274.33 \pm 413.21$ | $13655.05 \pm 5288.29$ | $574.32 \pm 811.43$ | $\mathbf{46734.79 \pm 30468.85}$ |
| Pong | $-15.02 \pm 1.96$ | $\mathbf{20.29 \pm 0.65}$ | $17.91 \pm 6.61$ | $19.31 \pm 2.42$ |
| Private Eye | $5727.66 \pm 1619.10$ | $79347.98 \pm 29315.82$ | $64145.31 \pm 21106.93$ | $\mathbf{100798.90 \pm 1.07}$ |
| Q*BERT | $3806.99 \pm 1654.33$ | $\mathbf{437607.43 \pm 111087.57}$ | $96189.83 \pm 17377.23$ | $238453.50 \pm 272386.91$ |
| Riverraid | $6077.47 \pm 1810.98$ | $56276.56 \pm 6593.69$ | $40266.92 \pm 4087.60$ | $\mathbf{90333.12 \pm 4694.40}$ |
| Road Runner | $25303.07 \pm 4360.56$ | $168665.40 \pm 40390.00$ | $\mathbf{447833.33 \pm 128698.32}$ | $399511.83 \pm 111036.59$ |
| Robotank | $13.67 \pm 2.55$ | $\mathbf{116.93 \pm 10.64}$ | $87.79 \pm 5.85$ | $114.46 \pm 3.71$ |
| Seaquest | $2146.63 \pm 1574.51$ | $\mathbf{999063.77 \pm 1160.16}$ | $577162.47 \pm 56947.06$ | $960181.39 \pm 25453.79$ |
| Skiing | $-25261.49 \pm 1193.77$ | $-4289.49 \pm 628.37$ | $-3401.56 \pm 185.93$ | $\mathbf{-3273.43 \pm 4.67}$ |
| Solaris | $2968.25 \pm 1470.62$ | $\mathbf{39844.08 \pm 6788.17}$ | $13514.80 \pm 1231.17$ | $28175.53 \pm 4859.26$ |
| Space Invaders | $640.13 \pm 185.45$ | $35150.40 \pm 3388.53$ | $33214.80 \pm 5372.10$ | $\mathbf{57828.45 \pm 7551.63}$ |
| Stargunner | $11214.14 \pm 4667.13$ | $\mathbf{796115.29 \pm 73384.04}$ | $221215.33 \pm 13974.19$ | $264286.33 \pm 10019.21$ |
| Surround | $-8.57 \pm 0.69$ | $8.83 \pm 0.58$ | $9.64 \pm 0.17$ | $\mathbf{9.82 \pm 0.05}$ |
| Tennis | $-18.34 \pm 2.41$ | $\mathbf{23.40 \pm 0.15}$ | $23.18 \pm 0.53$ | $22.79 \pm 0.65$ |
| Time Pilot | $3561.51 \pm 1114.00$ | $382111.86 \pm 17388.79$ | $169812.33 \pm 37012.23$ | $\mathbf{404751.67 \pm 17305.23}$ |
| Tutankham | $106.68 \pm 13.87$ | $\mathbf{2012.54 \pm 2853.44}$ | $402.16 \pm 22.73$ | $1030.27 \pm 11.88$ |
| Up'n Down | $15986.44 \pm 2213.66$ | $\mathbf{614068.80 \pm 32336.64}$ | $472283.82 \pm 23901.66$ | $524631.00 \pm 20108.60$ |
| Venture | $477.71 \pm 251.24$ | $2544.90 \pm 403.53$ | $2261.17 \pm 66.39$ | $\mathbf{2859.83 \pm 195.14}$ |
| Video Pinball | $18042.46 \pm 2773.10$ | $\mathbf{885718.05 \pm 54583.24}$ | $778530.78 \pm 79425.86$ | $617640.95 \pm 127005.48$ |
| Wizard Of Wor | $3402.48 \pm 1210.12$ | $\mathbf{134441.09 \pm 8913.57}$ | $67072.67 \pm 13768.12$ | $71942.00 \pm 6552.86$ |
| Yars' Revenge | $26310.18 \pm 6442.63$ | $\mathbf{976142.42 \pm 3219.52}$ | $654338.02 \pm 100597.12$ | $633867.66 \pm 128824.41$ |
| Zaxxon | $7323.83 \pm 1819.10$ | $\mathbf{195043.97 \pm 18131.20}$ | $79120.00 \pm 9783.55$ | $77942.17 \pm 6614.61$ |

Table 6: Scores per game when training and evaluating with *sticky actions* (Machado et al., 2018).

| Game | MEME @ 200M | MEME @ 1B | Go-Explore |
|---|---|---|---|
| Alien | 48076.48 ± 10310.65 | **68634.82 ± 15653.10** | |
| Amidar | 7280.27 ± 808.09 | **20776.93 ± 4859.39** | |
| Assault | 27838.75 ± 4337.41 | **31708.64 ± 14199.48** | |
| Asterix | **843493.60 ± 126291.56** | 729820.40 ± 82360.83 | |
| Asteroids | 212460.60 ± 5585.36 | **335137.50 ± 32384.14** | |
| Atlantis | 1462275.60 ± 144898.14 | **1622960.80 ± 1958.79** | |
| Bank Heist | 6448.48 ± 3066.16 | **45019.92 ± 8611.39** | |
| Battle Zone | 756298.00 ± 71092.41 | **763666.00 ± 53978.21** | |
| Beam Rider | **54395.06 ± 8299.92** | 38049.60 ± 3714.79 | |
| Berzerk | 16265.88 ± 10497.83 | **45729.94 ± 13228.29** | 197376 (@10B) |
| Bowling | **264.73 ± 0.89** | 212.70 ± 65.93 | 260 (@10B) |
| Boxing | 99.77 ± 0.10 | **99.86 ± 0.11** | |
| Breakout | **521.32 ± 49.04** | 475.87 ± 53.73 | |
| Centipede | 55068.43 ± 6370.11 | 63792.64 ± 24203.69 | **1422628** (@10B) |
| Chopper Command | 170450.00 ± 318026.00 | **181573.80 ± 342149.46** | |
| Crazy Climber | 272227.40 ± 24884.40 | **291033.20 ± 5966.79** | |
| Defender | 521330.30 ± 10194.48 | **561521.30 ± 7955.85** | |
| Demon Attack | 130545.44 ± 9343.72 | **142393.12 ± 1119.30** | |
| Double Dunk | 23.12 ± 0.58 | **23.78 ± 0.09** | |
| Enduro | 2339.31 ± 13.75 | **2352.94 ± 14.33** | |
| Fishing Derby | 68.12 ± 4.92 | **79.65 ± 2.65** | |
| Freeway | 33.88 ± 0.08 | 33.92 ± 0.02 | **34** (@10B) |
| Frostbite | 137638.50 ± 38943.68 | **498640.46 ± 38753.40** | |
| Gopher | **105836.08 ± 13458.22** | 96034.76 ± 17422.13 | |
| Gravitar | 12864.10 ± 260.14 | **19489.40 ± 825.38** | 7588 (@10B) |
| H.E.R.O. | 27998.94 ± 5920.99 | **175258.87 ± 15772.55** | |
| Ice Hockey | 38.14 ± 9.23 | **52.51 ± 10.04** | |
| Jamesbond | **144864.80 ± 8709.86** | 118539.20 ± 47454.71 | |
| Kangaroo | 15559.00 ± 1494.83 | **16951.60 ± 275.12** | |
| Krull | **127340.42 ± 27604.33** | 93638.02 ± 28863.77 | |
| Kung-Fu Master | 182036.20 ± 4195.48 | **208166.80 ± 2714.94** | |
| Montezuma's Revenge | 2890.40 ± 42.06 | 9429.20 ± 1485.32 | **43791** (@10B) |
| Ms. Pac-Man | 25158.68 ± 1786.43 | **27054.73 ± 152.14** | |
| Name This Game | 29029.20 ± 2237.35 | **34471.30 ± 2135.99** | |
| Phoenix | 440354.08 ± 70760.72 | **788107.78 ± 33424.51** | |
| Pitfall! | 235.90 ± 493.77 | **7820.94 ± 16815.61** | 6954 (@10B) |
| Pong | 19.38 ± 0.58 | **20.71 ± 0.08** | |
| Private Eye | 90596.05 ± 19942.76 | **100775.10 ± 15.57** | 95756 (@10B) |
| Q*BERT | 137998.15 ± 86896.43 | **328686.85 ± 257052.72** | |
| Riverraid | 36680.36 ± 1287.15 | **67631.40 ± 4517.53** | |
| Road Runner | 515838.00 ± 153908.19 | **543316.20 ± 64169.67** | |
| Robotank | 93.47 ± 4.18 | **114.60 ± 4.61** | |
| Seaquest | 474164.86 ± 62059.73 | **744392.88 ± 41259.26** | |
| Skiing | −3339.21 ± 14.59 | **−3305.77 ± 8.09** | −3660 (@10B) |
| Solaris | 13124.24 ± 657.30 | **28386.28 ± 2381.29** | 19671 (@20B) |
| Space Invaders | 26243.09 ± 6053.57 | **52254.64 ± 4421.24** | |
| Stargunner | 173677.60 ± 19678.82 | **190235.40 ± 6141.42** | |
| Surround | 9.60 ± 0.24 | **9.66 ± 0.23** | |
| Tennis | **22.65 ± 0.73** | 22.61 ± 0.53 | |
| Time Pilot | 159728.20 ± 39442.90 | **354559.80 ± 22172.76** | |
| Tutankham | 383.85 ± 61.43 | **924.47 ± 130.59** | |
| Up'n Down | 478535.54 ± 15969.44 | **528786.12 ± 5200.79** | |
| Venture | 2318.20 ± 56.39 | **2583.40 ± 175.95** | 2281 (@10B) |
| Video Pinball | 750858.98 ± 115759.28 | **759284.69 ± 37920.13** | |
| Wizard Of Wor | 65005.80 ± 7034.75 | **66627.00 ± 9196.92** | |
| Yars' Revenge | **654251.90 ± 121823.94** | 556157.86 ± 147800.84 | |
| Zaxxon | **85322.00 ± 12413.86** | 69809.20 ± 2229.49 | |

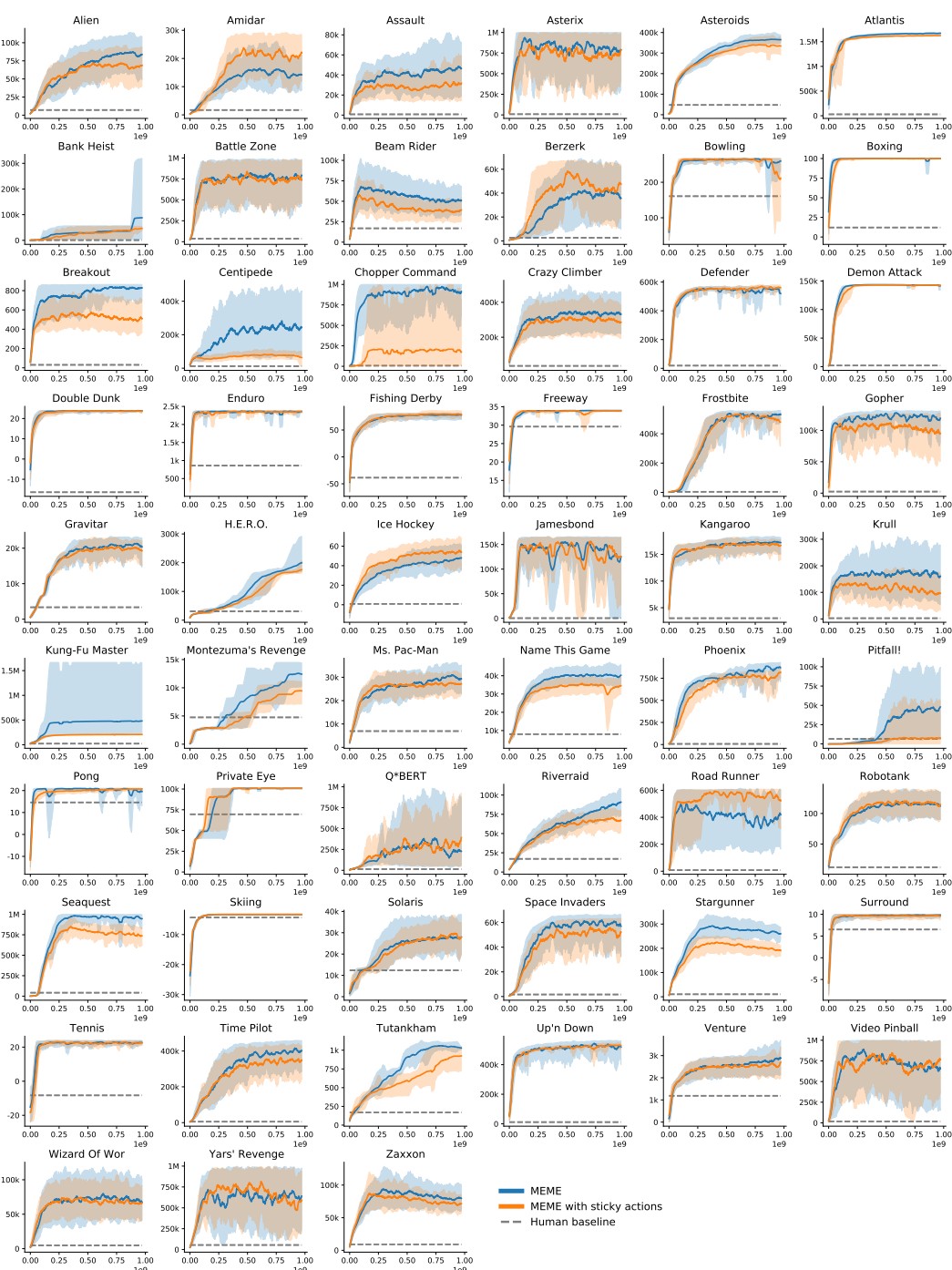

Figure 26: Score per game as a function of the number of environment frames, both with and without *sticky actions* (Machado et al., 2018). Shading shows maximum and minimum over 6 runs, while dark lines indicate the mean.

