# OpenReview forum: "Human-level Atari 200x faster"
_ICLR.cc/2023/Conference — ICLR 2023 poster_

### Official Review · Reviewer_SNik · 2022-10-17

**Confidence:** 5
**Correctness:** 2
**Technical Novelty And Significance:** 1
**Empirical Novelty And Significance:** 2
**Recommendation:** 3

**Clarity, Quality, Novelty And Reproducibility:**

Clarity: good
Quality: poor
Novelty: poor
Reproducibility: No code release is expected, but ample amount of information (including choices of hyper params) is given.

**Details Of Ethics Concerns:**

Too much energy consumption for too little result. I can only imagine how many test runs had to be conducted (hence, how many computational hours) before reaching a point to report the outcome. I raised a flag here mostly for the AC and senior AC to looking into such research practices and whether or not they should be considered as acceptable at ICLR today.

**Strength And Weaknesses:**

Strength:

- An obvious improvement over the Agent57.

- Perhaps the only interesting part of the paper is the trust region for the value function.

Weaknesses:

- Atari is simply a benchmark and not a goal. I honestly do not see any merit in these types of so-called research, especially when considering the tremendous amount of carbon emission and un-necessary energy consumption.

- Most of the presented techniques are ad-hoc with little formal analysis and reasoning. This hardly adds to the common knowledge of the community, and in light of the demanding computational requirements in the development stage of such projects, it is unlikely to open up new avenues for research either.

- Approximate GPU hours for the entire project not reported. As "responsible research" has become an important part of the AI community at large, such information helps the community to evaluate the merit of this work aside from possible hypes.


**Summary Of The Paper:**

The submission introduces an agent called MEME, which is built on Agent57, which in turn is a super inefficient atari player that requires a ridiculous number of training samples, not even remotely being sensible. Numerous ad-hoc or semi ad-hoc techniques are used (most of which with little to no formal ground) to mitigate various issues of Agent57. The resulting agent (MEME) is able to reduce the training samples by a factor of 200.

**Summary Of The Review:**

The paper provides several different modifications to Agent57, nearly none of which is formally studied or justified. As a reader, when I went through different sections, they only raised the question of "why". What is clear is that this manuscript is far from a research paper.

---

> ### Author Response · Authors · 2022-11-16
> **Response to Reviewer SNik**
>
> We thank reviewer SNik for their comments. On each one of the weaknesses:
>
> - Atari is indeed a benchmark and attaining human-level performance, efficiently, has been a goal of part of our community for the last 10 years [1, 2, 3, 4]. Atari was made to measure progress on the capacity of agents to tackle a wide range of tasks. In this paper, we demonstrate drastic improvements in data efficiency of a base agent while maintaining general performance–the very focus of this paper is to improve data efficiency and thus reduce compute usage by future research.
> - On the justification of the methods presented. 1) Each advancement is introduced with clear justification and basis in the broader literature. Each technique is then empirically evaluated. Theoretically derived results are not guaranteed to work empirically, nor are empirically excellent results always easily justified by theory. 2) Even though it is out of the scope of this publication, most techniques that are presented here could be applied to other agents, with encouraging results that are shown on applying such improvements to R2D2 as a base agent.
> - We do have a compute usage description on Appendix G, where we detail compute in terms of TPUv4 (TPUv4 specs are publicly available), as well as an indication of the speed this translates into in terms of actor and learner steps per second. In addition, in our response to reviewer tLk9, we have provided above an indication of the difference in wallclock time between Agent57 and MEME.
>
> On energy consumption: many of the success stories and peer-reviewed work of recent years are indeed energy intensive. This paper does precisely constitute a step in the direction of reducing energy consumption. In this case, the massive reduction in data used also corresponds to a reduction in compute time, and we also used more energy efficient hardware (with TPUv4 being on average 1.74x more energy efficient than A100 on our backbone ResNet architecture [5]).
>
> [1] Lake et al., Building Machines That Learn and Think Like People (https://arxiv.org/abs/1604.00289)
>
> [2] He et al., Learning to Play in a Day: Faster Deep Reinforcement Learning by Optimality Tightening (https://arxiv.org/abs/1611.01606)
>
> [3] van Hasselt et al., When to use parametric models in reinforcement learning? (https://arxiv.org/abs/1906.05243)
>
> [4] Schmidt & Schmied, Fast and Data-Efficient Training of Rainbow: an Experimental Study on Atari (https://arxiv.org/abs/2111.10247)
>
> [5] https://cloud.google.com/blog/products/ai-machine-learning/google-wins-mlperf-benchmarks-with-tpu-v4

---

> > ### Comment · Reviewer_SNik · 2022-12-06
> > **Thanks for the Response**
> >
> > I thank the authors for their response. The main point of my comments remains valid --- trying various tricks on one example does not lead into extendable knowledge and understanding. Formal analysis is not just proving bounds or convergence of an algorithm. It's rather about establishing the reasoning why something is expected to work and when. My assessment of this submission remains the same.

---

### Official Review · Reviewer_tLk9 · 2022-10-23

**Confidence:** 4
**Correctness:** 4
**Technical Novelty And Significance:** 3
**Empirical Novelty And Significance:** 3
**Recommendation:** 8

**Clarity, Quality, Novelty And Reproducibility:**

This paper is clear and easy to follow. The details about hyperparameters for reproducing are dedicatedly listed in the appendix; thus, it seems to be reproduced, especially, with good robustness shown in experiments.

200x in Title is a little tricky, since it does not reach 200x for all Atari games? It would be safe to say that in the abstract. I suggest modifying the title.


**Strength And Weaknesses:**

**Strength:**
The quality of writing and survey are good, and the result is state-of-the-art in terms of training speedups at human level. Data efficiency is a major problem in practical DRL cases; the proposed methods exhibit good results.
Since this paper has shown great improvement in the aspect of the game frame, I would also like to know how fast it is if we use wall time for counting the speedup (like many distributed methods focused).

**Weaknesses:**
It is an unclear part to me about the Retrace update: does MEME use transformed Bellman operator? (There is a “Value function rescaling” in the appendix-A)
Would the transformed value function be related to the part “B” (handling different value scales) or not? I do not see any mention of the transformed value function in both the main paper or appendix.

A minor comment is about references: it is better to cite papers to conferences, instead of arXiv as listed as follows.
- Distributed prioritized experience replay => ICLR 2018
- Image augmentation is all you need: Regularizing deep reinforcement learning from pixels => ICLR 2021
- Fast and Data Efficient Reinforcement Learning from Pixels via Non-parametric Value Approximation => AAAI 2022
- Prioritized experience replay => ICLR 2016
- Data-Efficient Reinforcement Learning with Self-Predictive Representations => ICLR 2021
- V-MPO: On-Policy Maximum a Posteriori Policy Optimization for Discrete and Continuous Control => ICLR 2020
- CURL: Contrastive Unsupervised Representations for Reinforcement Learning => ICML 2020


**Summary Of The Paper:**

This paper improves Agent57 in the following four aspects, faster propagation of learning signals, handling different value scales, efficient NN architecture, robustness about policy update. It achieves great speedup on training human-level Atari AIs, and 200-fold reduction.
The ablation experiments are complete and careful. Also, the authors provide very detailed information in the appendix.


**Summary Of The Review:**

Overall, this paper is good, providing several improvement techniques for Agent57 with significant speedups. Also, these techniques are well studied with ablations.

---

> ### Author Response · Authors · 2022-11-16
> **Response to Reviewer tLk9**
>
> We would like to thank the reviewer for the constructive feedback. Please see below our responses to each comment.
>
> > Does MEME use the transformed Bellman operator?
>
> We do still make use of the value function transform used in R2D2 and Agent57. The value transform primarily serves to compress the range of value function scales that need to be represented, but does not specifically address the problem of learning multiple value functions simultaneously. We had tested if it was possible to remove the value transform after introducing our TD-normalization scheme, and while the agent did remain stable, it still had a clear negative impact on performance across the Atari suite. A possible explanation for this is that the running statistics will take some time to adapt when there is a large sudden change in value scale, which could temporarily result in large gradients; but more work is needed to fully understand this phenomenon. The used value-transformation function is described in Table 2 (Appendix A). We updated the paper to include a reference to the value transform in the background section for clarity.
>
> > 200x in Title is a little tricky
>
> We agree that there was some potential ambiguity in the text which might result in the 200x figure being misinterpreted. The sense in which the 200x speedup is meant is in the number of frames required for _all_ games to surpass the human benchmark, rather than a claim about speedup in individual games. Specifically, with Agent57 the slowest game to reach human-level performance is Skiing (78B frames), while with MEME it is Pitfall (390M frames). The ratio of these two numbers is 200 which is the speedup we report. We’ve added a detailed clarification about this in section 5.1 and reworded the abstract to remove some ambiguity.
>
> > I would also like to know how fast it is if we use wall time for counting the speedup
>
> Ballpark figures for wall clock time of MEME using the resources detailed in Appendix G is slightly less than 1 day, vs 2 weeks for the published Agent57 results. But we note that the number and type of resources used in each of these works is different so this is not strictly a fair comparison.
>
> > It is better to cite papers to conferences, instead of arXiv
>
> Thanks for pointing this out. We have replaced arXiv references with conference ones in the updated version of the manuscript.

---

### Official Review · Reviewer_zSus · 2022-10-25

**Confidence:** 3
**Correctness:** 4
**Technical Novelty And Significance:** 3
**Empirical Novelty And Significance:** 4
**Recommendation:** 8

**Clarity, Quality, Novelty And Reproducibility:**

The paper is well written, the results are novel and the information necessary for reproducing the results is provided in the appendices.

**Strength And Weaknesses:**

Strengths:
- Identify a range of issues and bottlenecks in the original Agent57 design and systematically address them in addition to any side effects of their interventions, providing a set of tools that could be applied to a variety of problems along with an evaluation of their effectiveness.
- The result of these proposed methods is strong, at a 200 fold improvement in sample efficiency and competitive performance to MuZero and Muesli.
- Very thorough ablations, evaluation of methods on R2D2 in addition to Agent57, and an extended appendix including a list of things the authors tried but didn’t work, facilitating future research.

Weaknesses:
- This is a strong paper which claims impressive improvements on sample efficiency against Agent57 and supports those claims with thorough experiments.

**Summary Of The Paper:**

This paper introduces MEME, a method built off of Agent57 that enables exceeding the human baseline results with 200x fewer samples than Agent57 required.  This is accomplished by identifying bottlenecks and proposing solutions along with strategies to ensure stability and robustness.  The paper includes thorough experiments demonstrating the benefits of the proposed method and ablations providing intuition about each of the components.

**Summary Of The Review:**

This paper had a clear and significant goal, dramatically improving the sample efficiency of Agent57, which it states concisely and accomplishes through a wide range of well motivated modifications, as demonstrated through extensive experimentation.  I recommend acceptance.

---

> ### Author Response · Authors · 2022-11-16
> **Response to Reviewer zSus**
>
> We would like to thank the reviewer for the positive feedback. We have uploaded an updated version of the manuscript with additional seeds for the ablations in the main paper and fixing a bug in Appendix M in the R2D2 experiments.

---

### Author Response · Authors · 2022-11-16
**Updates to the manuscript**

We would like to thank all reviewers for their feedback. We have uploaded a new version of the manuscript that includes the following changes:

- We have doubled the number of seeds for all ablations presented in the main text.
- The results of Appendix M regarding R2D2 were affected by a bug and we have rerun these experiments to correct the issue, though we note that the qualitative differences in performance remain largely unchanged.
- We added some background on the value-function transform in section 2.
- We added an additional clarification in section 5.1 to explain where the 200x speedup reported in the title comes from and reworded the abstract to remove some ambiguity.
- We have updated the references to cite the conference version of papers instead of arxiv preprints.

---

### Decision · Program_Chairs · 2023-01-20

**Decision:**

Accept: poster

**Justification For Why Not Higher Score:**

The experiments are limited to Atari games.

**Justification For Why Not Lower Score:**

The experimental results are impressive.

**Metareview: Summary, Strengths And Weaknesses:**

This paper introduces MEME, a method built off of Agent57 that enables exceeding the human baseline results with 200x fewer samples than Agent57 required. This is accomplished by combining techniques. Extensive experiments are also provided.
The AC agrees with reviewer SNik that Atari is a benchmark and not the goal. On the other hand, Atari is a widely used benchmark in reinforcement learning, and chasing the benchmarks is one of the drivers for recent advances in machine learning. This paper shows by combing various (mostly known) techniques, it is possible to improve the sample efficiency significantly and achieve the state-of-the-art result. This experimental result itself is useful information to the community. The AC thus recommends acceptance.

**Note From Pc:**

if the above contains the word "oral" or "spotlight" please see: "oral" presentation means -> notable-top-5% and "spotlight" means -> notable-top-25%. As stated in our emails, we are disassociating presentation type from AC recommendations